# Negotiating science funding: The interplay of merit, bias, and administrative discretion in grant allocation in Kazakhstan

**Darkhan Medeuov**[1]☯*, **Kamilla Rodionova**[2], **Zhaxylyk Sabitov**[2,3], **Adil Rodionov**[4,5]☯

1 Department of Sociology and Anthropology, School of Sciences and Humanities, Nazarbayev University, Astana, Kazakhstan, 2 L.N. Gumilyov Eurasian National University, Astana, Kazakhstan, 3 Research Institute for Jochi Ulus Studies, Astana, Kazakhstan, 4 Institute of Eurasian Integration, Astana, Kazakhstan, 5 Maqsut Narikbayev Institute for Network and Development, Maqsut Narikbayev University, Astana, Kazakhstan

☯ These authors contributed equally to this work.

* darkhan.medeuov.personal@gmail.com

**Data availability statement:** Anonymized data and replication code are available at https://doi.org/10.7910/DVN/8UZ7EY.

## Abstract

This paper analyzes 4,488 applications from a grant funding competition held in 2017 in Kazakhstan. The competition had a two-stage design: first, anonymous subject matter experts evaluated the applications' scientific potential; then, open panels of local science managers made the final decisions. We analyze a range of bibliometric, institutional, and demographic variables associated with the applications and show that review scores account for only a small variation in success rates. The most important factor is the organizational closeness to decision-making. Gender also plays a role: we find that, net of academic merit, men and women investigators receive similar review scores, yet the panelists grant awards to men more often than to women. We further demonstrate that the gender gap emerges due to decisions made in a specific domain—Natural Resource Management.

## Introduction

Public research grants are among the most institutionalized methods of investing in science [1], but determining who deserves funding is a complex task. While merit is certainly an important criterion [2], economic, social, and political concerns also influence funding decisions. Scientists often value autonomy and believe that their internal rankings are the best predictors of future scientific and societal impact [2,3]. However, because science frequently relies on public funds, external actors (e.g., funding bodies, science administrators) introduce their own perspectives and agendas. The ways in which scientists and administrators negotiate and reconcile their demands present an intriguing sociological question.

A typical grant competition consists of two stages: first, grant applications are reviewed and scored by subject matter experts; second, selection panels decide whether a project deserves funding. Research identifies many factors influencing positive reviews or grant awards, including scientific merit, evaluative cultures, administrative discretion,

**Funding:** This research was funded by the Science Committee of the Ministry of Education and Science of the Republic of Kazakhstan [grant number AP13068350]. The funders had no role in study design, data collection and analysis, the decision to publish, or the preparation of the manuscript.

**Competing interests:** The authors have declared that no competing interests exist.

previous success, and, unfortunately, gender and race biases [2,4–9]. However, fewer studies have explicitly addressed the direct interplay between review scores and funding decisions [4,5].

We contribute to the discussion by analyzing the 2017 grant competition held by the Ministry of Education and Science of Kazakhstan that followed this typical two-stage design. Our data consist of 4,488 grant applications with extensive information on principle investigators' academic track records, institutional affiliations, and demographics. We specifically analyze the effect that peer review scores had on the final panel decisions across seven possible funding sections.

The present study adds to the literature in two respects. First, it extends the "geography of data." Most research on grant distribution comes from the US or Western Europe, contexts that arguably share some institutional similarities. A "developing" country, however, may present a comparative case that reveals different "logics" of grant distribution. In Kazakhstan, for example, official state discourse routinely urges scientific communities to meet the "doing science" standards of "advanced" societies. From the official perspective, a hallmark of those standards is commercialization[1]. This "catching-up" mindset can create specific institutional pressures on national science funding bodies, mapping of which may offer insights into the global division of labor in knowledge production systems.

The second contribution comes from analyzing the interaction between peer reviews and panelists' decisions. By design, peer reviewers were predominantly international scholars— subject matter experts commissioned to evaluate the scientific potential of applications (e.g., novelty, feasibility). In contrast, panelists were representatives of local academic and industrial institutions, forming the local academic establishment. The final decision rested with the panels, which considered review scores but had significant discretion. We find that, in some domains, review scores were more predictive of success than in others. We argue that in theoretical disciplines, panelists and experts share a common understanding of what constitutes "good" science. In contrast, panelists are less compliant in areas of applied science, where expertise is more tacit. This discrepancy arises, we argue, from the varying cognitive distance between experts and administrators [10].

The article is organized as follows: in the next section, we review the literature and sketch the grant funding system in Kazakhstan. Then we present the data and basic descriptive statistics. A 3-step data analysis follows: (a) we model peer-review scores using OLS regression, (b) we model the probability of getting funded using logistic regression; and (c) we cross-validate and compare logistic models with some fine-tuned random forest models. In the last section, we articulate the main takeaways and discuss the limitations.

## Background and theory

Grant awarding bodies face a selection problem: determining which projects deserve funding [1]. One common, if not normative, solution is to let the community decide. From this perspective, scientists tend to see their peers as suitable juries because internal rankings within a discipline are considered a reliable measure of future scientific output. Peer recognition is seen as a means to ensure valid scientific knowledge because radical openness to peer criticism [11] and "organized skepticism" [12] help filter out unwarranted or dubious claims. Pragmatically, merit-based selection is also appealing to scientists because it implies field autonomy [3,13]. Indeed, research indicates that selection committees, at least in principle, take meritocracy seriously and believe that it should guide their decisions [2].

---

[1]   For example, recently the Ministry of Education and Science has organized an international forum on commercialization in science, https://www.gov.kz/memleket/entities/sci/press/news/details/862596?lang=ru.

One common way to infer scientific merit is peer review, a process in which grant applications are scored by multiple, often anonymous, subject matter experts [1]. The practice has long been institutionalized in sciences and almost all large funding bodies (e.g. NIH, European Research Council) base their decisions on some sort of peer review basis [14].

In the meantime, peer review has a long history of criticism [15,16]. Some studies show that positive peer reviews do not guarantee measurable scientific impact in publications and citations [17]. Others raise concerns that since one bad review can be enough to block an application, peer reviews prioritize safe, conventional avenues of research at the expense of high-risk, high-reward proposals [18]. Relatedly, commentators point out that peer reviews lack consistency: reviewers often disagree on scientific merit and weight of evaluation criteria, and inter-reviewers consistency is typically low [14,19–22]. The way the information on projects is presented also seems to sway reviewer scores as the NSF experiment suggests [23].

Partly, peer reviews can appear inconsistent due to factors such as reviewer fatigue, vague evaluation criteria (e.g., significance or novelty), or a lack of reviewing experience [14,24]. However, disagreements between reviewers also stem from the inherent ambiguity of merit. Scientific communities may agree on what constitutes quality science and who among them comes closest to doing it, yet the literature indicates that consensus is not always reached. Many scientific fields have controversial topics that split communities into what are often called schools of thought [12,25]. While the degree of (perceived) fragmentation vary across disciplines [25–27], sociological studies of scientific knowledge show that the nature of scientific fact can be, and often is, contested even in the most stable of hard sciences like physics or chemistry [28–31].

The realization that scientists occasionally disagree even on basic principles led some to suggest that peer reviews should be used only to filter out "poor" proposals, while the final decision should be based on a closer, consensual panel discussion. And indeed, many grant funding agencies follow two-step selection procedures where, after the initial anonymous peer review, panels of experts collegially reach final decisions (e.g., NIH). Panels tend to be more diverse than peers: they may contain outsiders to a discipline and supposedly offer a more holistic view of a project's merit. For example, panelists with industry experience may better understand a project's economic impact [1].

Empirical studies, however, show that open panels do not eliminate inconsistencies and still face the same basic challenges as peer reviews. Variation in success rates suggests that panelists have cognitive and organizational biases; and studies into the discourse of decision-making show that consensus is often tentative, situational, and guided by heuristics, personal tastes, and strategic considerations besides formal criteria of merit [2].

In part, increasing uncertainty is the obverse of a holistic perspective. Funding bodies often have to go beyond merit alone and reckon with societal, political, and economic considerations. This is visible , for example, in ensuring equal funding chances for women and men, particularly in STEM research [4]. With this in mind, many funding bodies (e.g. NIH) explicitly retain discretion in decision-making, so the administrators can fund a project "out-of-order" if it aligns with their agenda, or compensate for institutional barriers that some social groups may face, as, for example, Bol and colleagues show in the case of the Netherlands' grant funding system [4].

This brings us to other demographic and institutional factors that may correlate with peer reviews or success probability, net of scientific merit. Studies identify a wide range of such sources: gender, institutional affiliation, educational background, co-publication record, grantsmanship, race, and ethnicity [4–9]. For example, affiliation and academic pedigree can

be interpreted by reviewers and panelists as additional indicators of merit or be taken into strategic consideration if, for example, reviewers or panelists share affiliations with the applicant. In the slightly different context of manuscript peer review, Teplitsky and colleagues find that review scores correlate with the closeness in the co-authorship network [25], suggesting that reasons for favorable reviews and decisions can be a result of sharing the same epistemic cultures [32] or belonging to the same school of thought [12].

Moreover, composing a successful application, or grantsmanship, is a separate skill that requires experience and understanding of the selection process. Previous success, then, can easily accumulate, creating the so-called "Matthew effect" [6,12]. Finally, it is crucial to recognize that non-meritocratic factors, such as an applicant's gender, ethnicity, or race, can significantly impact the decision-making process. For example, gender often serves as a primary characteristic for homophily in scientific collaborations, meaning that individuals tend to form partnerships and networks with those of the same gender. As a consequence, not only collaboration networks but also citation networks show systematic gender homophily [33,34]. Research also shows systematic penalties for women in both hiring and grant funding [4,8]. There is also evidence that minority groups can have inexplicably lower success rates, net of other merit and institutional factors, as Ginther and colleagues showed in the NIH grant distribution for Asian and Black investigators [7].

The upshot of this section of the literature review is that both peer reviews and panel decisions are shaped not only by the scientific validity, significance, or novelty of a proposal, but also by a battery of institutional, cognitive, and even affectual biases of referees, complicated further by administrative discretion. The selection process is asymmetric, and while decision-makers take peer reviews into account, they have considerable leeway in their final decisions. As we mentioned earlier, panels look beyond scientific validity or novelty, so peer reviews, in principle, should account only for a part of variation in success rates [6]. A question, however, remains what can mediate the accord between peer reviews and panel evaluations?

From a normative perspective, peer reviewing and panel evaluation are essentially forward projections of future successes [25]. Both peers and panelists attempt to assess what impact a scientific work will have. The aspects of impact they are scoping, however, differ. Very crudely, peers predict success in the community of scientists (e.g., citations), while panelists predict, among other things, societal impact. For example, when choosing between equally quality-assuring projects in astro- and geophysics, panelists may find the latter easier to justify on the grounds of public benefit. In this regard, a study of the Netherlands' funding program reveals an interesting, though indirect, insight. Bol and colleagues [6] show that applications "around the funding cutoff" diverge over time in the amount of grant money secured but show no significant difference in the number of citations received. So grants, in that case, begot more grants, but not citations.

The implication for our context is that panelists, when not having clear excellence signals (review scores are about the same), may try to reduce uncertainty by looking at an investigator's track record. Extending this line of logic, we might argue that any uncertainty makes panelists either infer "grantworthiness" from other proxy measures or take advantage of uncertainty by rewarding themselves (self-service) or colleagues (organizational proximity) [35,36]. Additionally, uncertainty allows for biases to enter into consideration. For example,
Bagues and colleagues argue that gender starts playing a role only in the absence of other merit signals [8].

The situation above applies to a small subset of applications in a narrow window around the "cutoff threshold." However, a signaling framework allows extending the scope to the whole distribution of scores. We may ask to what degree, in principle, external experts

are authoritative for panelists? This may depend on the extent panelists can understand technical evaluation and where they see themselves within a discipline's internal hierarchy. In other words, how big is the cognitive distance between experts and administrators, and what do administrators think of their own technical expertise?

Scientific disciplines vary in the extent to which they agree on preferred endeavors, theories, methods, or tools. Some disciplines feature institutional structures that impose a higher degree of homogeneity on the education and training required for students to begin their scientific careers [37,38]. Others disciplines exhibit greater diversity in their epistemic styles [2]. We argue that the level of epistemic diversity may correlate with the stability of evaluation criteria. Specifically, disciplines with more homogenous epistemic styles may tend to have more fixed sets of criteria for distributing resources (e.g., grants, academic positions). This suggests that in fields with low epistemic diversity, science managers may be more inclined to rely on external experts, whose authority helps reduce uncertainty (and potentially shift responsibility). Conversely, disciplines characterized by more diverse epistemic cultures often, by having more flexible evaluation criteria, allow administrators to exercise greater discretion in their decisions.

In our context, it is important to note that all the participating projects were grouped into seven rather broadly defined categories, allowing, for instance, projects in physical cosmology to compete directly with those in differential geometry, or projects in sociology with those in history. The competition occurred in the sense that a single panel of managers was responsible for deciding on funding for all types of projects within one domain. This suggests that panels, by aggregating projects from diverse fields, could be described as epistemically diverse in terms of the range of disciplines represented. However, the degree of epistemic diversity within the panels likely varied depending on how closely related the grouped fields were and how much methodological or theoretical overlap existed between them. For instance, a panel overseeing both physical cosmology and differential geometry might exhibit less epistemic diversity than one spanning sociology and history, despite both being composed of projects from distinct disciplines.

To conclude, we analyze interactions between panelists and reviewers as a signaling problem. Panelists receive signals from reviewers, and when they deem those signals unclear, they infer "grantworthiness" using proxy measures—potentially allowing biases to influence their decisions—or by opportunistically exploiting uncertainty (e.g., for self-reward or organizational proximity). The extent to which panelists comply with reviewer scores may vary depending on the epistemic diversity of the domains. Arguably, in domains comprised of "hard" sciences (e.g., Mathematics, Physics), experts' opinions may be taken more seriously than in more discursive fields.

In the next section, we will describe the Kazakhstani case and specify how findings from the literature may apply to its context.

## Materials and methods

### Context

In 2017 the Ministry of Education and Science of Kazakhstan held a call for grant applications. 4488 proposals were submitted, of which 1097 were awarded 3-year funding (success rate ~24%). The 2017 call was the third one in a sequence of 3-year funding programmes started in 2011. Previously, the Ministry of Education and Science (the main funding body) had no specific investigator-oriented program and distributed funds in an ad hoc manner. The process was mostly top-down, no peer review was involved, and administrators had full discretion in the selection. The probability of getting a grant at that stage arguably depended on investigators' connections with the ministry. In 2011, however, the law "On Science" No.

407-IV was passed. This law, among other things, instituted two important requirements: obligatory external, anonymous peer review and national research councils, collegial bodies of elected domain experts who were bestowed rights to make decisions on funding.

The grant competition became eligible for different groups of scholars, with no restrictions on disciplines, career stages, or workplace (an applicant does not have to be affiliated with a university or a research organization). The amount of financial support was set at 180 million Tenge ($\sim$ 500 thousand USD) for the natural and technical sciences and up to 90 million Tenge ($\sim$250 thousand USD) for the social sciences and humanities.

The 2017 call was organized as follows. In the first stage, researchers submitted applications to the National Center of Science and Technology Evaluation (NCSTE), a body under the Ministry of Education and Science. NCSTE's task was to check compliance with formal requirements and find three anonymous external domain experts to evaluate applications. Each expert scored the application on 0 to 36 points (4 criteria on 0 to 9 scale), and the final application's score was the mean of the experts' scores. NCSTE paid experts for each review. The exact amount of remuneration has not been disclosed as well as the selection criteria for the reviewers. Regulating documents outlined only general principles: reviewers must be active and impactful scientists in their fields and their h-index must be above five over the last five years (so they had to publish at least five papers, each had been cited at least 5 times over the last five years). Also, at least two of the reviewers must be international scholars. This last requirement was especially important for the Ministry to legitimize the whole procedure.

Not much had been clarified on how experts and projects were matched either. Regulations say nothing specific on this front, yet some anecdotal evidence suggests that the NCSTE was regularly in touch with certain field experts. Basically, administrators were calling experts, showed them lists of applications, and allowed experts to choose projects to evaluate.

After review scores were received, the NCSTE passed the applications and experts' reviews to the national scientific councils (NSCs). Panels of council members would assemble and discuss projects in their priority order (high score projects first). After the discussion, panelists would vote anonymously. Success was determined by a majority vote. Councils did not have to explain their decisions.

The NCSTE, the organization formally responsible for organizing the call, had no jurisdiction over councils' decisions. The latter were collegial bodies, independent of the Ministry of Science and Education on paper; their members were proposed by academic and industry communities. The catch, however, was that the Ministry had the jurisdiction to approve the candidates, and the principles upon which approval was based, again, were not explicitly stated. In fact, the regulations were so general, and the list of organizations eligible to supply council members was so inclusive that practically any government official had a formal chance of getting on board. Perhaps the direct excerpt from the law would illustrate this ambiguity the best,

> Compositions of councils shall be formed by the authorized body from among competent Kazakhstani and foreign scientists, representatives of state bodies, national managing holdings, national development institutes, national holdings, national companies, subjects of private entrepreneurship on proposals and recommendations of sectoral authorized bodies, scientific organizations, organizations of higher and (or) postgraduate education, scientific public associations, other organizations and shall be approved in accordance with. ("On Science" No. 407-IV)

To be precise, regulations set some scientometric criteria for the members. At least 50% of a council must be practicing scientists with 1) at least ten years of experience in a discipline

associated with the council's domain, 2) h-index above three, and 3) at least five papers within the last five years published in Q1-Q3 journals (Scopus or Web of Science). And for the Social Sciences and Humanities domain, the h-index requirement was relaxed.

So scientific councils were composed of both scientists and nominees of governmental bodies. Those latter did not necessarily have a relevant scientific background. For example, some of the council members were representatives of the National Chamber of Commerce. On top of it, as public commentators observed later, the actual composition often diverged from the formal requirements.[2]

However, the most interesting part about the councils was that some of their members were actually applying for funding. Essentially, it was up to research council members to decide whether they deserved money or not[3].

Unsurprisingly, the 2017 grant application call resulted in a scandal. The decisions made by the councils, their dubious compositions, and the fact that some of the council members applied for grants themselves became focal points of backlash from the applicants (unsuccessful and successful alike), as well from some prominent journalists. When the Ministry released the winners list, the applicants realized that a high expert score did not guarantee success: some projects that scored high did not receive grants, while those with mediocre scores did. A group of frustrated scientists even signed a collective letter and released a public video, asking then-President Nazarbayev (!) himself to look into what they called a blatant case of corruption in the science system of Kazakhstan[4]. The ministry didn't stand idle either; it issued a series of self-vindicating press releases[5,6] and even tried to sue (unsuccessfully) one of the signatories on the charges of spreading misinformation[7].

In the next section, we present quantitative characteristics of that notorious application call and show to what degree alleged nepotism, among other factors, correlates with peer review scores and success likelihood.

## Data

The initial data set consists of 4,488 grant applications and contains projects' titles, short descriptions, the principal investigator's identification, and peer review scores. We used the principal investigators' identifications to manually collect information on their academic metrics (e.g., h-index, Scopus-listed publications) and institutional affiliations. Below, we list all the variables we used with a short description of their type and how we constructed them.

Some of the explanatory variables had a fraction of missing data. We were unable to locate data on the academic degrees of 185 PIs (about 4%), most of whom were from the field of agriculture. We have imputed missing values using random forest imputation with default parameters (number of trees = 10) implemented in the MICE package [39].

**Academic merit.  Peer review score** (numerical): Each application had been graded by three independent anonymous experts based on four criteria. Each criteria ran on 0 to 9 scale, where 9 is the best grade, and consisted of 2-3 categories with a set of cue questions. The categories were

---

[2]    The ministry's spokesperson disagreed with that in an interview. We haven't evaluated whether their claims were correct or no, but we speculate that the disagreement might be due to the interpretative flexibility of the selection criteria.

[3]    And sometimes they decided that they didn't.

[4]    https://www.youtube.com/watch?v=67TRlE_wsCU&t=1s.

[5]    https://informburo.kz/novosti/skandalnoe-raspredelenie-grantov-na-nauchnye-proekty-prokommentirovali-v-mon-rk.html.

[6]    https://informburo.kz/stati/kto-vhodit-v-nacionalnye-nauchnye-sovety-i-kak-raspredelyayut-granty-dlya-uchyonyh.html.

[7]    https://informburo.kz/novosti/skandal-s-raspredeleniem-nauchnyh-grantov-nesoglasnymi-uchyonymi-zanyalas-policiya.html.

- novelty and actuality (0-9);
- quality and realizability (0-9);
- significance of the expected results (0-9);
- competence and scientific track record of the research group (0-9).

The cue questions varied in their ambiguity. Some were more or less concrete, like

How high is the standard of journals selected for publishing research results? (*novelty and actuality*)

or

How well planned are the experiments for the subsequent statistical treatment of the data obtained? (*quality and realizability*)

Others held potential for interpretations,

What are the possible social, economic, environmental or other effects of the project? (*significance of the expected results*)

or

How likely is it that articles published as a result of the project will be regularly used and cited?" (*significance of the expected results*)

The total score from an expert was the sum of the category subscores. The total application score was the mean of the experts' scores. It is exactly this mean expert score that we take as a variable. In an ideal scenario, having a full breakdown of review scores would be more informative, as it would allow for the modeling of disagreement among reviewers as a fixed effect.[8] However, the public data published by the Ministry contained only the mean score.

One grading category, *competence and scientific track record of the research group*, was specifically asking experts to evaluate the project teams' research capacity with questions like "do project members appear to be qualified enough for the job?". This implies that reviewers might infer competence from publication or citation counts potentially making review scores correlate with formal measures of merit such as h-index.

Also, the grading rubric did not distinguish between different research domains, meaning that projects in the domains of *Science*, *Culture*, or *National security* were evaluated on the same criteria and sets of cue questions.

Besides review scores, we collect information on investigators' bibliometric profiles, variables include:

**Scopus** (binary): This variable indicates whether a PI had published in journals listed in the Scopus database (by the end of 2018). While tracing just one Scopus publication may be obsolete in other national contexts, Kazakhstani academia is still mostly Russian-speaking and a Scopus-listed publication is not yet a universal characteristic: about 30% of PIs have not had any publications indexed in Scopus (N = 1342).

**h-index** (numeric): a PI's Hirsch Index according to Scopus by the end of 2018.

**Russian Science Citation Index** (binary): RSCI is analogous to Scopus or Web of Science but for publications in Russian. RSCI is important, because of the strong historical connections between post-Soviet academic institutions. In some fields, like mathematics, publications in RSCI journals may count towards an academic degree and be a proxy of academic merit.

---

8    We thank one of the anonymous reviewers for pointing out this possibility.

**Scopus-delisted** (binary): A Scopus based publication became a condition for getting an academic degree somewhere in 2011. Arguably, that requirement increased incentives for local research to publish in "predatory" journals. While "predatoriness" of journals can be hard to measure or even conceptualize [40], there are some heuristics to establish a lower bound of a journal's quality. Scopus regularly delists journals that do not meet their quality criteria. This variable indicates if a PI had ever published in journals that were removed from the Scopus database.

**Variables measuring grantsmanship skills (Matthew effect).   Previous success** (binary). This variable indicates whether a PI received funding in the previous grant competition held in 2014. Although winning the previous competition likely correlates with a PI's academic fitness, this variable can also approximate a PI's experience with the selection process or signal "grant-worthiness" to the panelists in the absence of other cues. Ideally, we would like to have information on the full grant track record of participants, including all types of funding they received. In the absence of such data, we interpret this variable as a conservative indicator of previous experience, as it only captures those who participated in the previous round of the grant competition (thus increasing the rate of false negatives).

**PI's formal degree** (*categorical*, 3 levels). This variable has three categories: *Candidate of Science* (reference level), *Doctor of Science*, and *PhD*. Before joining the Bologna Process in 2011, Kazakhstan followed the Soviet two-tier system of scientific degrees: *Candidate of Science* (first tier) and *Doctor of Science* (second tier). Since the 2000s, this system has been replaced by the single *PhD* degree. Because we do not have data on the year of degree award, this variable may capture cohort effects across different generations of academics. It likely reflects variation in the accumulation of social capital throughout participants' careers, as well as differences in institutional experience, both of which may influence the likelihood of funding.

**Variables measuring institutional capital.   Closeness to the decision-making process** (*categorical*, 3 levels). This is one of the main explanatory variables. Formally, members of the NSCs could submit their proposals for the call on an equal basis with other applicants, which inevitably created a conflict of interest. To address this issue, committee members were required to leave the meeting when their applications were discussed (a similar approach is described in [41]). However, self-exclusion arguably did not fully resolve the problem, as committee members might still be inclined to support each other's applications, perhaps under the assumption that their colleagues would reciprocate.

To measure indirect connection to decision-makers, we added an intermediary category - "works with" (following approach from [36]) - for those PIs that share affiliations with the council members. This is a rough proxy for a social or institutional closeness to council members. We assume that semiformal social networks of friendship and patron-client relationships span the research community, and, ideally, we would like to have some sort of "who is friends with whom" information. In its absence, however, the best we can do is to control for shared affiliations and assume correlation with a higher probability of being in some sort of semiformal relationship

**Organizational scale** (categorical, 4 levels). Each project had to identify an "operating organization" (e.g., a university, research institute, museum) that would manage the funds on behalf of the project. Typically, the operating organization is the institution with which the PI is affiliated. We hypothesize that affiliation with a larger organization (e.g., a national university) may influence funding decisions. In addition, organization scale is an important control variable for inferring the impact of proximity to decision-makers, as we measured it as shared affiliation with a research council member. Council members tend to work in larger organizations, so to separate the effect of organization from the effect of proximity,

we control for the size of organizations. We divide the organizations in which project members work into 4 categories: Regional (baseline), National, International, and the Rest. This division is somewhat subjective and reflects some vague consensus about the size of the university student body. The "Rest" category includes all organizations that we could not categorize as either regional, national, or international; most of them actually are research institutes formerly associated with Kazakh SSR Academy of Science. In the period of independence, after numerous reforms, the academy was transformed into a public organization, losing direct state funding, as well as the prestige and importance of the main hearth of "high science" [42].

**Other variables.  Domain** (categorical, 7 levels). Grant applications had to select one of the seven research domains. Below are their short descriptions. The shortcut names that we use in tables and graphs are in parentheses.

1. Fundamental and applied research in the field of social sciences and humanities (*Culture*, baseline category);
2. Rational use of natural resources (*Natural RM*);
3. Energy and mechanical engineering (*Energy*);
4. Information, telecommunication and space technologies, research in natural sciences (*Science)*;
5. Life and Health sciences (*Life*);
6. Sustainable development of agro-industrial complex and safety of agricultural products (*Agriculture*);
7. National security and defense without secrecy (*Security*).

For the reference category, we chose *Culture*, because it had the largest number of applications.

**PI's gender** (binary, men is the reference category). We use the term *gender* not in the sense of self-identification but rather to describe how individuals were categorized within the register system used to collect applications. We acknowledge that the use of *gender* instead of *sex* is debatable, as the system apparently referred to the *sex* assigned on identification documents (e.g., national passports). Our argument for using *gender*, however, is that it was likely the panelists' and reviewers' perception of applicants' *sex* that influenced their decisions. In that sense, since this perception was socially conditioned and tied to the identification system imposed by state institutions, we assume that *gender* is a more appropriate term.

A PI's gender is an important variable to account for, as research documents controversial evidence of gender biases in academia. For example, McAlister and colleagues report that "Women make up 33% of the applicants who are eligible for programmes funded by the UK Biotechnology and Biological Sciences Research Council, but they lead only 21% of grant applications" [43]; Burns and colleagues find that women had a significantly lower success rate in applications submitted to the Canadian Institutes of Health Research [44]. Using a natural experiment data from the same institution, Witteman and colleagues also document that women success rate was lower than that of men by a significant margin, age and research domain adjusted [45]. On the other hand, a study of the Australian Science Fund program found no gender gap in review scores [20] and a study of grant in the Social Science department at the University of Hong Kong find even slight women advantage in grant award success rates [46]. This mixed evidence, indeed, suggests that contexts matters, and examining the case of Kazakhstan may contribute towards building a processual and situational understanding of gender in academia.

Beyond direct discrimination, gender can also influence outcomes through *homophily* [33, 34] understood as the preference for similarity [12]. Gender-based homophily could potentially act as a selection mechanism, especially given the gender composition of research councils. While women researchers accounted for 46% of PIs, the research councils were predominantly male (about 71%).

**Region of the operating organization** (categorical, 4 levels). Geographic localization of the operating organization. Approximately 70% of projects were from organizations located in the two largest cities: Almaty and Astana. We also singled out Shymkent as the third largest city. Other regions were grouped into the "Other" category. Almaty was selected as the base category.

**Project rank** (categorical, 4 levels): a PI could lead at most two projects, so the number of unique PI identities (3889) is less than the total number of projects (4488): 601 PIs (∼15%) submitted two projects. We account for the project's ranking by categorizing it into the following four groups: *only* (if a PI has only one project), *best* (when the project has a highest score), *second* (when scored second), and *tie* (when both projects received the same score). The base-line level is *only*. Ideally, we would like to have fixed effect models for investigators, yet that would require having more applications over the same investigators (and most of our investigators submitted only one application). As an interim solution we cluster standard errors by PIs for both OLS and logit models.

Our analysis consists of three stages. First, we model experts' review scores using OLS regression. Then, we model the probability of winning using logistic regression. Finally, we checked predictive performance with 10-fold validation and compared our best logistic regression models with a fine-tuned random forest on several common metrics.

## Results

### Modeling review scores

As mentioned above, the Kazakhstani grant program includes seven domains. Table 1 summarizes the number of applications, success rates, typical scores, mean and median scores, and other statistics across the domains. Fig 1 displays success rates for different score intervals by domains. Scores, in general, correlate with success rates, yet the strength of association seems to vary across domains.

To understand variation in review scores, we run OLS regressions by domains. We exclude data from the Security domain because of the small number of applications (N = 112). Before we proceed, however, we need to address a few methodological issues.

We treat the variable *review score* here as a continuous one, while in fact it can take on only a finite number of values. More over, conceptually it is more sound to treat *score* as an ordinal variable, as its values reflect levels of academic fitness assessed by the reviewers (akin

**Table 1. Descriptive statistics for review scores and success rates across domains.**

| Domain | N | Winners | Success rate | Mean score | Median score | SD | Q1 | Q3 |
|---|---|---|---|---|---|---|---|---|
| Culture | 1304 | 236 | 18.10 | 23.31 | 23.67 | 4.92 | 20.00 | 27.00 |
| Agriculture | 583 | 79 | 13.55 | 23.89 | 24.00 | 4.12 | 21.00 | 27.00 |
| Science | 565 | 169 | 29.91 | 25.01 | 25.33 | 4.77 | 21.67 | 28.67 |
| Life | 561 | 204 | 36.36 | 23.80 | 24.00 | 4.86 | 20.33 | 27.67 |
| Security | 112 | 25 | 22.32 | 23.83 | 23.67 | 4.75 | 21.00 | 27.33 |
| Natural RM | 1045 | 306 | 29.28 | 24.53 | 25.00 | 4.24 | 21.67 | 27.67 |
| Energy | 326 | 84 | 25.77 | 24.41 | 24.84 | 4.88 | 20.67 | 28.33 |

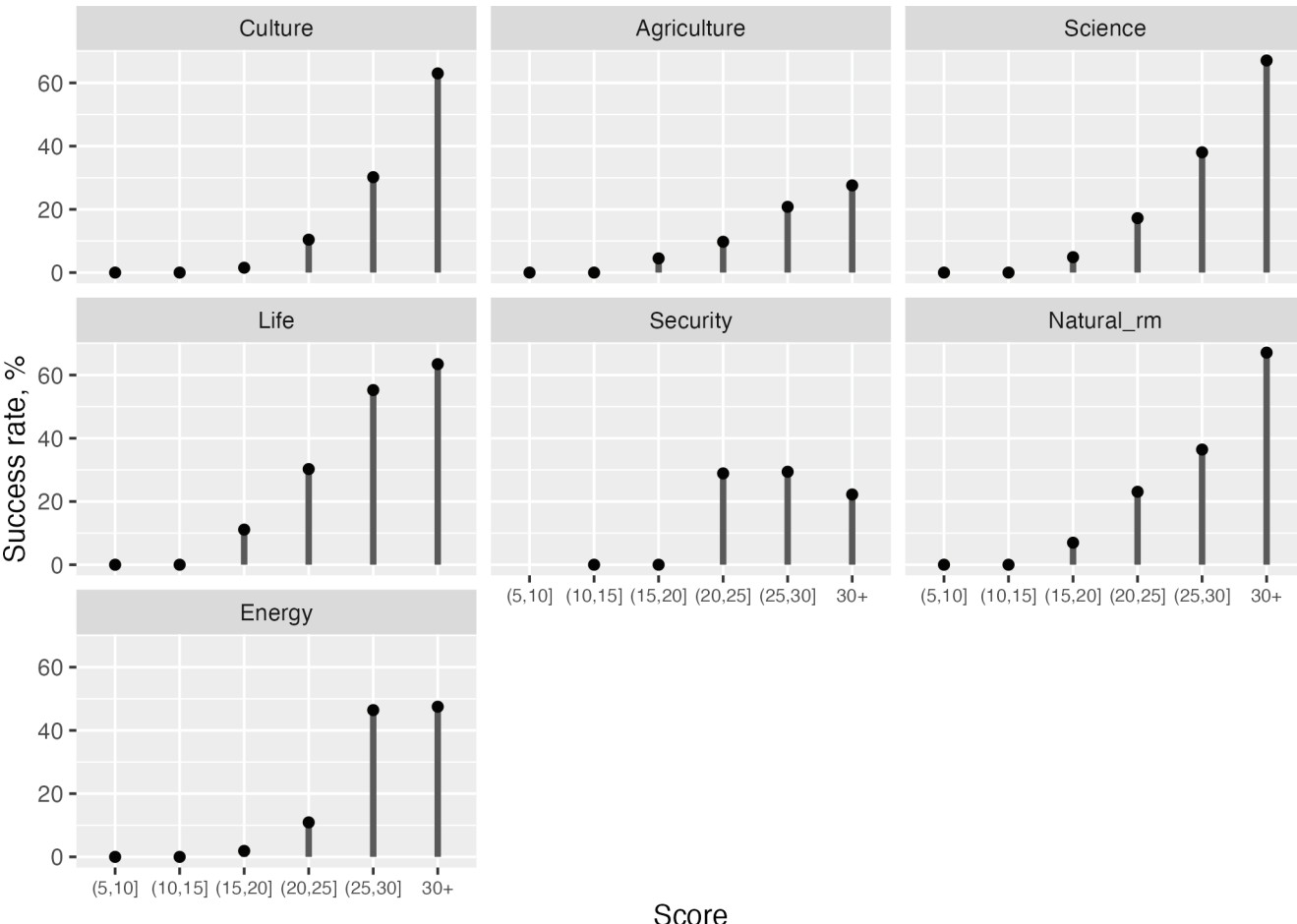

**Fig 1. Success rates by score intervals and domains.** Applications are grouped in review score intervals. Success rates shows what proportion of applications within a given interval received funding.

to IELTS or TOEFL scores). A more principled approach would be to model *reivew score* with an ordinal regression treating its values as ordered levels.

This paper, however, mainly analyses the effect of scores on the probability of getting a grant. In other words, *review score* is an explanatory variable, and our OLS regressions do not explain it, but rather describe it in relation to other variables. So as not to sidetrack too much, we organize the analysis as follows. Below, in Table 2, we present the results of OLS regressions that treat *review score* as a continuous variable. In the supporting information (S2 Table, S3 Table), we run model diagnostics and alternative models that treat *score* as a discrete and ordinal variable.

First, we may observe that there is no one set of variables that explain variance in scores across all domains. Previous success matters almost everywhere (in Natural RM and Energy significance is lower). H-index also correlates with review scores everywhere, except for Culture. However, it should be noted that the evaluation criteria directly asked reviewers to assess the scientific capacity of the project teams. This part was likely evaluated by looking at PI's track record, among other things. So, the significance of the h-index could be induced by the evaluation design.

**Table 2. OLS regression with robust standard errors for review scores by domains.**

| | Science | Energy | Natural_rm | Culture | Life | Agriculture |
|---|---|---|---|---|---|---|
| (Intercept) | 21.028*** | 22.088*** | 23.152*** | 22.234*** | 22.193*** | 24.052*** |
| | (0.881) | (0.777) | (0.506) | (0.483) | (0.858) | (0.587) |
| gender:Female | 0.468 | 0.237 | −0.188 | 0.117 | −0.628 | 0.041 |
| | (0.405) | (0.678) | (0.266) | (0.283) | (0.397) | (0.360) |
| region:Astana | 0.401 | −1.302+ | −0.541 | −0.073 | 0.978+ | 0.422 |
| | (0.493) | (0.662) | (0.403) | (0.348) | (0.537) | (0.498) |
| region:Shymkent | −0.633 | −3.276** | 0.610 | −2.805*** | 1.037 | −0.486 |
| | (1.073) | (1.172) | (0.595) | (0.568) | (1.188) | (0.804) |
| region:Other | 0.212 | −1.142 | 0.067 | −0.200 | 0.615 | −0.782+ |
| | (0.764) | (0.809) | (0.392) | (0.428) | (0.681) | (0.452) |
| rints:Yes | 1.232** | 0.188 | 0.535+ | 0.823* | −0.072 | 0.929+ |
| | (0.444) | (0.646) | (0.325) | (0.372) | (0.737) | (0.483) |
| scopus:Yes | 1.131+ | 1.847** | −0.406 | 0.410 | −0.413 | −1.075* |
| | (0.675) | (0.689) | (0.339) | (0.380) | (0.449) | (0.426) |
| H-index | 0.298*** | 0.190* | 0.197*** | 0.207 | 0.228*** | 0.380** |
| | (0.047) | (0.090) | (0.060) | (0.177) | (0.065) | (0.118) |
| delisted:Yes | −2.610*** | −1.450* | 0.251 | −0.982** | −2.681*** | 0.401 |
| | (0.511) | (0.642) | (0.335) | (0.346) | (0.747) | (0.416) |
| Win 2014:yes | 1.945*** | 1.560** | 0.925+ | 1.763*** | 1.725*** | 1.358** |
| | (0.438) | (0.561) | (0.502) | (0.351) | (0.432) | (0.439) |
| degree:DoS* | 0.258 | 0.069 | 0.665* | 0.683* | −0.291 | 0.500 |
| | (0.418) | (0.548) | (0.281) | (0.283) | (0.425) | (0.393) |
| degree:PhD | 1.251* | 0.575 | 1.389*** | 0.549 | 0.252 | −0.630 |
| | (0.518) | (0.713) | (0.414) | (0.513) | (0.598) | (0.679) |
| Inst cap:Works with | −0.315 | 2.163*** | 1.051* | 1.325** | 1.119* | −0.045 |
| | (0.549) | (0.597) | (0.494) | (0.406) | (0.564) | (0.422) |
| Inst cap:Member | −2.672* | −1.330 | 0.120 | 0.867 | −0.860 | −1.402 |
| | (1.047) | (1.882) | (1.027) | (1.339) | (1.038) | (1.047) |
| Inst Cap:Missing | 0.222 | | −4.627** | −0.581 | −2.724 | |
| | (1.959) | | (1.518) | (0.970) | (2.418) | |
| Org scope:National | 1.727* | −1.010 | 0.218 | −0.346 | 1.007 | −0.919 |
| | (0.812) | (0.828) | (0.570) | (0.527) | (0.830) | (0.583) |
| Org scope:International | 2.710* | 6.422*** | 2.571 | 1.623 | −2.152 | |
| | (1.315) | (1.839) | (1.843) | (1.190) | (2.007) | |
| Org scope:Other | 1.774* | 1.367+ | 1.118** | 1.125** | 1.685* | 0.064 |
| | (0.702) | (0.758) | (0.396) | (0.408) | (0.699) | (0.494) |
| Num.Obs. | 565 | 326 | 1043 | 1299 | 561 | 582 |
| R2 | 0.252 | 0.236 | 0.072 | 0.111 | 0.162 | 0.061 |
| R2 Adj. | 0.228 | 0.196 | 0.057 | 0.099 | 0.136 | 0.036 |
| AIC | 3242.5 | 1905.4 | 5930.8 | 7704.4 | 3302.6 | 3296.5 |
| BIC | 3324.9 | 1973.6 | 6024.8 | 7802.6 | 3384.9 | 3370.8 |
| Log.Lik. | −1602.274 | −934.700 | −2946.388 | −3833.196 | −1632.318 | −1631.269 |
| RMSE | 4.12 | 4.26 | 4.08 | 4.63 | 4.44 | 3.99 |
| Std.Errors | HC1 | HC1 | HC1 | HC1 | HC1 | HC1 |

*DoS stands for Doctor of Science.

There are factors that do not show any significance in individual domains, for example, gender. When controlling for other academic and institutional variables, the projects led by women and men do not differ significantly in their average review scores. However, the gender gap becomes significant in the pooled data, likely due to the increase in statistical power.

Council members do not score higher than others, and in *Science* they score even less than the conditional mean by 2.7 points. However, investigators who work with council members score higher in the domains of Energy, Natural RM, Culture, and Life (with low significance though). Also, everywhere, except for Agriculture, affiliation with research institutions (the main component under the other rubric for organizational prestige) is associated with relatively higher scores.

The amount of explained variance changes over domains. In Science and Energy, the model captures around 23% and 20% of the variance, respectively; in Life and Culture 13.5% and 10%; in Natural resource management: about 7%; and in Agriculture: under 3%. These differences may reflect different evaluative conventions in the fields, yet it should be noted that explanatory variables do not vary much in certain domains, limiting models' capacity to explain variation in the response variable.

Next, we run some regression models on the total dataset. In particular, we examine differences between male and female investigators. A simple baseline model that models score as a function of gender shows a small, yet significant difference of ~0.7 points in received scores (Table 3, first column). This difference, however, disappears as more variables are controlled (Table 3, last column).

Using Gelbach decomposition [47], we are able to explain ~93% of the difference between male and female investigators. In particular, we find that about 28% of the difference can be attributed to h-index (mean male ~1.51, female ~0.8). H-index itself is heavily skewed, but the degree of skewness is different: ~54% and 61% of male and female investigators have 0 h-index, yet males have almost 3 times higher probability of having h-index above 5 compared with female investigators. Another major contributor to the score difference is the previous success: 20.8% of the male investigators won grant awards in the previous competition, vs. 12.9% for the females. Previous success accounts for about 20% of the score difference.

Other factors contributing to the difference are presented in Fig 2. Publishing in delisted journals, having a doctor of science degree, and working for a research institution together account for another 28% of the difference. In other words, women in this dataset were more likely to publish in journals delisted from Scopus (~31% vs. ~21% ), less likely to have a doctor of science degree (~31% vs. ~44%), and less likely to work in research institutions (~38% vs. ~43%).

## Modeling success

We now turn to our main response variable: winning grant funds. We model the probability of getting funded with a set of logistic regressions. Our prime explanatory variables are review scores, closeness to decision-making, and investigators' gender. These three variables presumably represent the degree to which academic, institutional, and gender concerns influence the council's decisions.

We start with a model that only accounts for review scores (Table 4, first column). While this model is unlikely to capture real decision-making, it is a useful baseline for comparing with other models. The next model looks into other merit related factors. At the third step, we include both review scores and other merit indicators. Then we specify a model with all available predictors except for the closeness to the decision process. And finally, we include all the available predictors at once. We exclude the closeness to decision making at the fourth step to show that this predictor is important both for overall model fit and the estimation of the effect of other variables.

First, review scores are robust to various model specifications: one unit increase in score is associated with ~1.26 increase in odds of getting a grant regardless of controls. Comparing

**Table 3. OLS regression for the whole data.**

| | sex | demo | demo+domain | demo+domain+delisted+qual | full | full, robust SE |
|---|---|---|---|---|---|---|
| (Intercept) | 24.346*** | 24.803*** | 24.038*** | 23.942*** | 22.656*** | 22.656*** |
| | (0.095) | (0.118) | (0.178) | (0.201) | (0.267) | (0.349) |
| sex:Female | −0.700*** | −0.682*** | −0.448** | −0.267+ | −0.057 | −0.057 |
| | (0.139) | (0.138) | (0.143) | (0.141) | (0.142) | (0.122) |
| region:Astana | | −0.238 | −0.068 | −0.165 | 0.015 | 0.015 |
| | | (0.175) | (0.177) | (0.174) | (0.177) | (0.235) |
| region:Shymkent | | −2.555*** | −2.462*** | −2.233*** | −1.338*** | −1.338 |
| | | (0.280) | (0.280) | (0.275) | (0.311) | (0.855) |
| region:Other | | −1.055*** | −0.934*** | −0.889*** | −0.361+ | −0.361 |
| | | (0.176) | (0.178) | (0.176) | (0.210) | (0.257) |
| domain:Agriculture | | | 0.472* | 0.309 | 0.237 | 0.237* |
| | | | (0.233) | (0.230) | (0.230) | (0.118) |
| domain:Science | | | 1.362*** | 0.395 | 0.175 | 0.175 |
| | | | (0.237) | (0.250) | (0.248) | (0.242) |
| domain:Life | | | 0.257 | −0.213 | −0.449+ | −0.449** |
| | | | (0.233) | (0.235) | (0.236) | (0.174) |
| domain:Security | | | 0.162 | 0.158 | 0.394 | 0.394** |
| | | | (0.456) | (0.449) | (0.444) | (0.138) |
| domain:Natural Rm | | | 1.086*** | 0.593** | 0.752*** | 0.752*** |
| | | | (0.195) | (0.198) | (0.198) | (0.113) |
| domain:Energy | | | 0.873** | 0.156 | 0.172 | 0.172 |
| | | | (0.291) | (0.292) | (0.289) | (0.123) |
| rints:Yes | | | | 0.782*** | 0.723*** | 0.723*** |
| | | | | (0.195) | (0.193) | (0.186) |
| scopus:Yes | | | | 0.088 | 0.028 | 0.028 |
| | | | | (0.172) | (0.170) | (0.305) |
| H-index | | | | 0.325*** | 0.262*** | 0.262*** |
| | | | | (0.029) | (0.029) | (0.036) |
| delisted:Yes | | | | −0.709*** | −0.645*** | −0.645 |
| | | | | (0.178) | (0.177) | (0.405) |
| Win 2014:yes | | | | | 1.597*** | 1.597*** |
| | | | | | (0.186) | (0.146) |
| degree:Doctor | | | | | 0.430** | 0.430** |
| | | | | | (0.148) | (0.166) |
| degree:PhD | | | | | 0.884*** | 0.884*** |
| | | | | | (0.223) | (0.226) |
| Inst cap:Works with | | | | | 0.759*** | 0.759* |
| | | | | | (0.173) | (0.306) |
| Inst cap:Member | | | | | −0.681 | −0.681 |
| | | | | | (0.526) | (0.550) |
| Inst Cap(Missing) | | | | | −1.729** | −1.729+ |
| | | | | | (0.634) | (1.042) |
| Org scope:National | | | | | −0.043 | −0.043 |
| | | | | | (0.242) | (0.319) |
| Org scope:International | | | | | 1.056 | 1.056 |
| | | | | | (0.669) | (0.818) |
| Org scope:Other | | | | | 0.830*** | 0.830** |
| | | | | | (0.197) | (0.308) |
| R2 Adj. | 0.005 | 0.027 | 0.037 | 0.074 | 0.103 | |
| AIC | 26,544.2 | 26,446.5 | 26,407.3 | 26,238.6 | 26,104.0 | 34,982.0 |
| RMSE | 4.65 | 4.60 | 4.57 | 4.48 | 4.41 | |

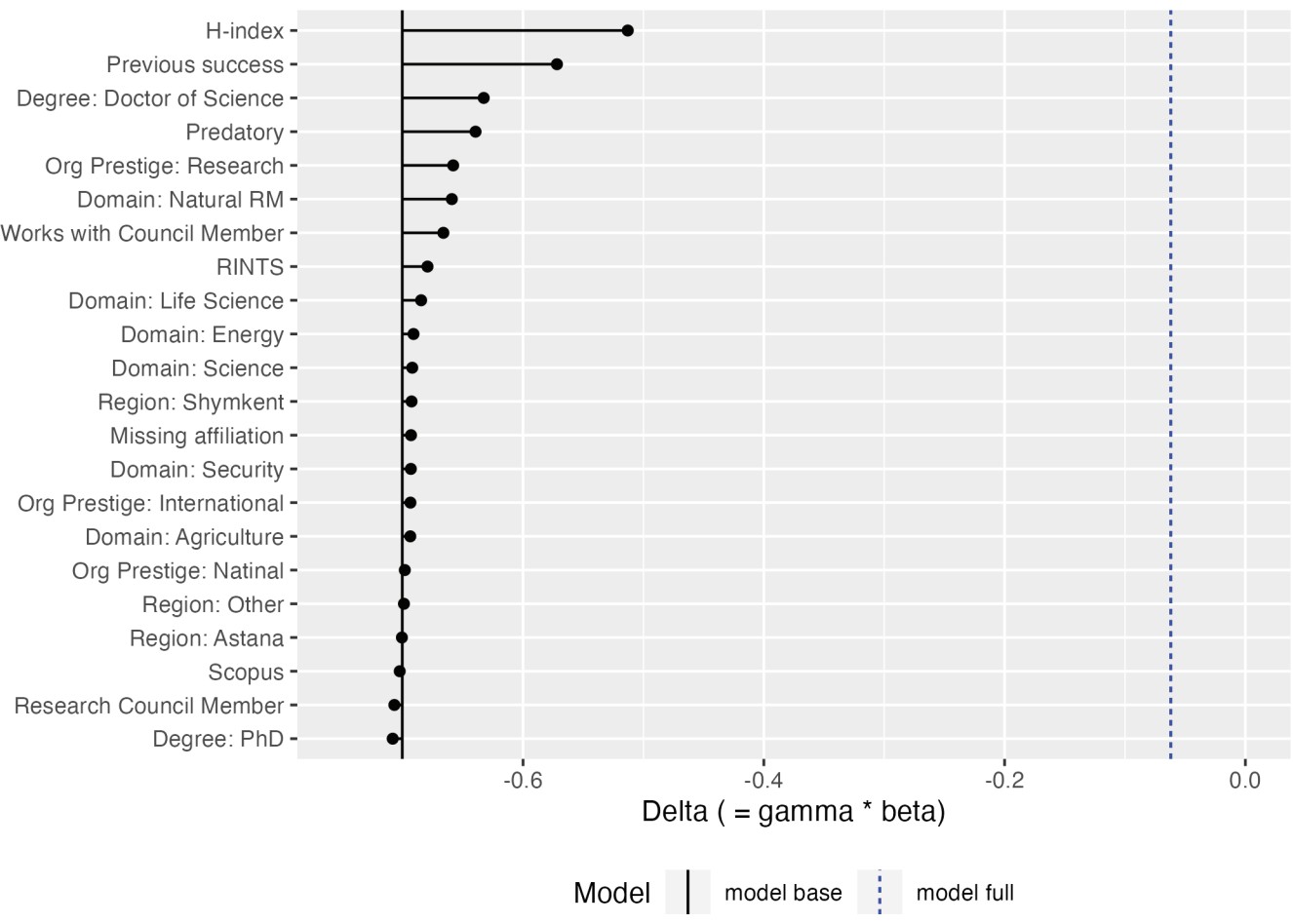

**Fig 2. Gelbach decomposition.** Variables' contribution to the gender gap in review scores. To decompose the gap, we used an R implementation by Matthieu Stigler [48].

merit related variables (the third column), we see that score, basically, subsumes other proxies of merit: h-index and publication profile decrease their statistical importance once review scores are taken into account. This squares well with an earlier observation that review scores are likely to be partially produced based on groups' scientific capacities.

Another thing to note is the decreasing importance of h-index. Besides scores, h-index correlates with domains (the highest being in Science) and membership in the research councils. When these variables are accounted for, h-index loses significance. For example, the council members top other investigators in h-index by more than 3 units (mean 1.12 vs. 4.48, median 0 vs. 2).

We also observe evidence for the "Matthew" effect (accumulated advantage): previous success increases odds by a factor of 1.7 ($e^{0.526}$), and having a doctor of science degree increases odds by 1.5 ($e^{0.439}$). A PhD degree, on the other hand, does not seem to give an edge compared with the reference category (candidate of science). Being a doctor of science works here as an indicator of "old school", since after 2011 Kazakhstan abolished the title. Doctors of science are likely to be older, and supposedly more authoritative scientists. Besides merit, their project may receive funding out of respect and/or fear of retribution.

**Table 4. Logistic regressions for winning a grant (with robust SE).**

| | merit | merit+memory+demo | full-inst_cap | full | full, scoreXsex | full, hirshXsex |
|---|---|---|---|---|---|---|
| (Intercept) | −7.323*** | −7.067*** | −7.277*** | −7.554*** | −7.441*** | −7.511*** |
| | (0.262) | (0.269) | (0.313) | (0.322) | (0.396) | (0.323) |
| Score | 0.246*** | 0.232*** | 0.225*** | 0.232*** | 0.228*** | 0.232*** |
| | (0.010) | (0.010) | (0.011) | (0.011) | (0.014) | (0.011) |
| H-index | | 0.075*** | 0.040* | 0.019 | 0.019 | 0.010 |
| | | (0.019) | (0.017) | (0.019) | (0.019) | (0.019) |
| rints:Yes | | 0.178+ | 0.158 | 0.121 | 0.120 | 0.124 |
| | | (0.106) | (0.112) | (0.115) | (0.115) | (0.115) |
| scopus:Yes | | 0.010 | 0.076 | 0.097 | 0.096 | 0.086 |
| | | (0.098) | (0.103) | (0.106) | (0.106) | (0.106) |
| delisted:Yes | | −0.190+ | −0.056 | −0.047 | −0.046 | −0.046 |
| | | (0.098) | (0.108) | (0.110) | (0.110) | (0.110) |
| Win 2014:Yes | | | 0.560*** | 0.516*** | 0.516*** | 0.518*** |
| | | | (0.103) | (0.105) | (0.105) | (0.105) |
| degree:Doctor | | | 0.454*** | 0.434*** | 0.434*** | 0.429*** |
| | | | (0.087) | (0.089) | (0.089) | (0.089) |
| degree:PhD | | | −0.073 | −0.146 | −0.146 | −0.147 |
| | | | (0.134) | (0.137) | (0.137) | (0.137) |
| sex:Female | | | −0.215* | −0.197* | −0.459 | −0.258** |
| | | | (0.086) | (0.088) | (0.549) | (0.098) |
| Pr Rank:Second | | | −0.675*** | −0.726*** | −0.728*** | −0.731*** |
| | | | (0.143) | (0.152) | (0.152) | (0.152) |
| Pr Rank:Best | | | −0.085 | −0.102 | −0.100 | −0.106 |
| | | | (0.113) | (0.114) | (0.114) | (0.114) |
| Pr Rank:Tie | | | 0.370 | 0.408 | 0.410 | 0.409 |
| | | | (0.547) | (0.543) | (0.540) | (0.542) |
| Inst cap:Works with | | | | 0.590*** | 0.589*** | 0.592*** |
| | | | | (0.107) | (0.107) | (0.107) |
| Inst cap:Member | | | | 3.088*** | 3.087*** | 3.088*** |
| | | | | (0.296) | (0.297) | (0.294) |
| Inst Cap:Missing | | | | 0.279 | 0.282 | 0.285 |
| | | | | (0.400) | (0.400) | (0.400) |
| Score×Female PI | | | | | 0.010 | |
| | | | | | (0.021) | |
| H-index×Female PI | | | | | | 0.052 |
| | | | | | | (0.045) |
| Organizational Fixed Effects | No | No | Yes | Yes | Yes | Yes |
| Regional Fixed Effects | No | No | Yes | Yes | Yes | Yes |
| Domain Fixed Effects | No | No | Yes | Yes | Yes | Yes |
| Num.Obs. | 4488 | 4488 | 4488 | 4488 | 4488 | 4488 |
| AIC | 4250.2 | 4218.3 | 4027.4 | 3910.0 | 3911.8 | 3910.1 |

Domains vary in success rates. Compared with *Culture* (the reference category), projects in the *Natural Resource Management* domain are 1.5 times more likely to be funded, and projects in *Life* (biomedical research) are almost 3 times more likely to be funded. This apparently reflects a) varying domain priorities and b) different entrance costs: *Culture* is the most populous domain probably because it is easier to draft a proposal that will qualify for the competition.

Now to address the elephant in the room. The most important predictor of winning is, unsurprisingly, just being a member of a research council. Checking that box increases the odds of receiving funding by a factor of 20. Working with a member also helps, sharing affiliation increases odds by 1.75 when other indicators of prestige and status are controlled.

Finally, we find evidence for gender discrimination: female investigators have, ceteris paribus, 20% less odds of winning a grant than their male counterparts. Here, we observe a situation almost opposite to that described in [4]. There, "gender-equal funding conceals unequal evaluations." In our case, gender-unequal funding conceals equal evaluations.

We also fit domain subsets separately and show that the aggregate effect of sex discrimination is driven by funding decisions coming from the Natural Resource Management. Table 5 details intradomain effects of the variables (excluding Security). Natural Resource Management is a domain that comes closest to parity between male and female investigators in terms of their sheer numbers, mean review scores, and mean h-index, yet women have less chances of getting money. To be precise, the gender gap in success rates exists in all domains except for Security. Its size varies from 3% (Energy) to 11% (Natural RM). Yet in other domains, this difference can be explained by the difference in review scores and h-index. Only in Natural Resource Management are men and women investigators are similar in their review scores and bibliometric output, yet their success rates still differ.

Before moving on, we would like to briefly address the issue of model selection. Our models yield many non-significant coefficients, yet we retain them. From a parsimony perspective, this is suboptimal, as it would make more sense to retain only those variables that possess greater explanatory power—and thus simplify the models.

The reason for retaining non-significant coefficients is primarily conceptual. Our explanatory variables were selected based on: (a) theoretical relevance and (b) data availability. Our main goal was to test whether these variables are associated with the probability of funding. When a variable is not (for example, the h-index), it still provides valuable information, as it arguably suggests that the review score already captures what the h-index measures.

However, to demonstrate the robustness of the coefficients, we fit several LASSO regressions [49] and compare their coefficients with those obtained in our final model (excluding interaction effects). The results are presented in the Supporting Information (S2 Figure, S5 Table).

## Cross-validation

Logistic regression is a well-established statistical method to model probability of a given event. However, in terms of predictive performance, logistic regressions do not always match with more recent non-parametric predictive models such as random forest. This predictive performance, however, comes at the cost of interpretation; non-parametric models do not have a closed-form equation connecting response and predictor variables. Nevertheless, comparing logit models with more flexible non-parametric models in terms of predictive performance helps us illustrate that in our case logistic models are on par with non-parametric ones.

We estimate the performance of the models using two 10-fold cross-validations. We choose three models from the logistic regression branch, one full model specification (with all available predictors), and two models with interaction effects (score X domain and score X sex). We also select two specifications for both random forest and XGBoost models. For those non-parametric models, we fine-tune parameters over a Latin hypercube grid to optimize the receiver operator area under the curve (roc auc) and logarithmic loss (log loss). S3 Figure and S4 Figure in the supporting information summarize our performance tests. The tests were done in Tidymodels R package [50].

**Table 5. Logit models for winning a grant by domains.**

|  | Science | Energy | Natural_rm | Culture | Life | Agriculture |
|---|---|---|---|---|---|---|
| (Intercept) | −8.707*** | −9.841*** | −6.181*** | −10.328*** | −6.063*** | −8.019*** |
|  | (1.017) | (1.317) | (0.654) | (0.710) | (0.767) | (1.187) |
| Score | 0.237*** | 0.295*** | 0.195*** | 0.335*** | 0.215*** | 0.224*** |
|  | (0.032) | (0.049) | (0.022) | (0.025) | (0.027) | (0.039) |
| gender:Woman | 0.070 | −0.334 | −0.417* | −0.243 | −0.090 | −0.123 |
|  | (0.260) | (0.433) | (0.162) | (0.185) | (0.214) | (0.297) |
| Win 2014:yes | 0.454+ | 0.231 | 0.587+ | 0.381+ | 0.536* | 0.878** |
|  | (0.255) | (0.376) | (0.310) | (0.219) | (0.244) | (0.314) |
| rints:Yes | −0.109 | 0.045 | 0.186 | 0.354 | 0.079 | 0.115 |
|  | (0.303) | (0.415) | (0.209) | (0.251) | (0.382) | (0.392) |
| scopus:Yes | 0.967* | 0.259 | 0.026 | 0.106 | −0.084 | 0.377 |
|  | (0.474) | (0.513) | (0.200) | (0.233) | (0.240) | (0.369) |
| H-index | −0.010 | 0.064 | 0.056 | 0.053 | 0.008 | −0.070 |
|  | (0.029) | (0.055) | (0.045) | (0.111) | (0.034) | (0.094) |
| delisted:Yes | 0.364 | 0.315 | −0.068 | −0.086 | −0.158 | 0.131 |
|  | (0.352) | (0.414) | (0.198) | (0.226) | (0.422) | (0.341) |
| region:Astana | −0.573+ | 0.053 | −0.092 | −0.426+ | 0.441 | −0.346 |
|  | (0.309) | (0.462) | (0.254) | (0.230) | (0.282) | (0.367) |
| region:Shymkent | −0.180 | 0.214 | 0.106 | −0.526 | −0.497 | −0.068 |
|  | (0.803) | (0.965) | (0.392) | (0.524) | (0.756) | (0.678) |
| region:Other | −1.111+ | 0.516 | −0.265 | 0.020 | −0.258 | −0.193 |
|  | (0.591) | (0.568) | (0.253) | (0.327) | (0.377) | (0.375) |
| degree:DoS | 0.971*** | 0.582 | 0.441** | 0.341+ | 0.208 | 0.183 |
|  | (0.280) | (0.363) | (0.169) | (0.189) | (0.230) | (0.286) |
| degree:PhD | −0.137 | −0.141 | −0.559* | 0.062 | 0.412 | −1.335* |
|  | (0.334) | (0.469) | (0.277) | (0.341) | (0.322) | (0.575) |
| Inst cap:Works with | 0.829* | −0.115 | 0.647* | 0.886** | −0.163 | 0.208 |
|  | (0.336) | (0.405) | (0.267) | (0.302) | (0.278) | (0.327) |
| Inst cap:Member | 2.109*** | 3.110*** | 3.262*** | 3.512*** | 17.686*** | 3.459*** |
|  | (0.557) | (0.750) | (0.719) | (0.891) | (0.545) | (0.727) |
| Inst Cap:Missing | −0.451 |  | −1.100 | 0.732 | 0.117 |  |
|  | (1.585) |  | (1.361) | (0.565) | (0.634) |  |
| Pr Rank:Second | −1.286** | 0.001 | −1.157*** | −0.323 | −1.070** | 0.386 |
|  | (0.457) | (0.444) | (0.290) | (0.340) | (0.348) | (0.483) |
| Pr Rank:Best | 0.466 | 0.069 | −0.114 | −0.206 | −0.346 | −0.013 |
|  | (0.291) | (0.445) | (0.223) | (0.276) | (0.310) | (0.355) |
| Pr Rank:Tie | 3.075* | −16.924*** | 0.051 | −1.061 | −16.786*** | 1.881 |
|  | (1.239) | (1.169) | (0.779) | (1.286) | (0.734) | (1.269) |
| Org scope:National | −0.564 | 0.681 | −0.059 | −0.386 | 0.052 | 0.114 |
|  | (0.436) | (0.585) | (0.327) | (0.410) | (0.440) | (0.455) |
| Org scole:International | 0.797 | −15.790*** | 1.322 | 0.760 | −15.749*** |  |
|  | (1.066) | (1.060) | (0.960) | (0.482) | (0.805) |  |
| Org scole:Other | 0.237 | 0.412 | 0.356 | −0.013 | 0.292 | −0.086 |
|  | (0.374) | (0.526) | (0.259) | (0.326) | (0.381) | (0.377) |
| Num.Obs. | 565 | 326 | 1043 | 1299 | 561 | 582 |
| AIC | 529.0 | 308.4 | 1086.7 | 910.0 | 616.2 | 417.9 |
| Std.Errors | HC0 | HC0 | HC0 | HC0 | HC0 | HC0 |

DoS stands for Doctor of Science. Domain *Security* is excluded due to the small number of observations.

## Conclusion

The study uses logistic regression to analyze the likelihood of obtaining a grant. Key explanatory variables include review scores, proximity to decision-making, and investigator gender, which represent the influence of academic, institutional, and gender factors on funding decisions.

The results show that review scores are consistently linked to a higher probability of funding: one unit increase in score increases the odds of winning by approximately 1.26. Other merit indicators lose significance once the review scores are considered. Previous success and holding a Doctor of Science degree also increase the odds of receiving funding.

The most significant predictor of winning a grant is being a member of the research council. Sharing affiliations with members also improves funding chances. Finally, gender bias is apparent, as female investigators have 20 percent lower odds compared with their male counterparts, net of other factors.

Another interesting result is that models, in fact, tend to underestimate the number of winners. For example, logistic regressions generally predict that only approximately 14% of projects would receive funding. And, more than half of the funded projects (56%) had the odds against them: they had low review scores, and their principal investigators did not distinguish themselves in terms of academic achievements, institutional capital, or connections with key decision-makers. And yet, they received funding.

Part of this "unexplained" success rate is likely due to unobserved project characteristics. For example, some projects might be funded simply because they were relatively inexpensive. In other words, council members might take the requested budget into account when deciding for or against a project. Unfortunately, we could not find publicly available data on project budgets.

We, however, argue that, net of other factors, research councils may seek to fulfill a certain number of grants. To a degree, they may use uncertainty to their advantage and distribute grants to themselves and organizationally proximate investigators; also, in some domains, decision makers show gender bias. However, in general, the councils seem to distribute money randomly. This may be due to fear of not using the allocated funds, which could lead to budget cuts in the next competition. This fear, often attributed to bureaucrats, fits into the picture of the bureaucratization of science, other signs of which, for example, focus on quantitative indicators of productivity such as publication count, are anecdotally abundant in local academia.

Based on the findings, several policy recommendations can be made to improve the efficiency of grant allocation. The first recommendation addresses gender inequality. It is crucial to determine whether this inequality is a recurring issue in grant competitions or specific to certain calls or domains. As our data suggest gender bias in the 2017 call was more pronounced in the domain of Natural Resource Management, mostly composed of projects related to oil industry.

If gender inequality proves to be a consistent problem across domains, implementing temporary quotas could help make up for biases. Previous research has shown that, while Kazakhstani academia has achieved quantitative gender parity — evidenced by the growing number of women obtaining scientific degrees — significant gender inequality persists [51]. In this context, such measures could be highly effective.

The second key policy recommendation concerns the members of the National Scientific Council (NSC). As decision-makers, they were also involved in the competition, which created a clear conflict of interest. The Ministry of Science and Higher Education has already taken steps to address this issue. In subsequent competitions, NSC members no longer have the final say in determining which projects receive funding; instead, they score projects alongside external reviewers. However, further in-depth research is needed to clarify the councils role, evaluate their contributions to the final decisions, and open the "black box" of communication between council members during the negotiations. For example, one mechanism that may contribute to gender bias is gender homophily coupled with disproportionate number of men in research councils. In other words, possible interventions may focus not on the

outcome of the competition ( for example, gender quotas for winning projects), but on gender composition of the councils (gender quotas for council members).

It would be misguided to assume that the issue with research councils is unique to Kazakhstan. Similar cases have been observed in European contexts, where selection committees sometimes disregard review scores [6] or exhibit self-serving bias [41]. Therefore, it is important not to exoticize the Kazakhstani case but to develop a unifying framework for addressing issues of gender inequality and the balance between external and internal expertise.

Another important discussion point is the extent to which our findings can be extrapolated to other academic contexts. First, it should be noted that one consequence of the 2017 grant call was a redesign of subsequent calls. In this sense, we cannot be certain that our findings are directly applicable to data from, for instance, the 2021 call. On the other hand, the gender bias patterns we observe are surprisingly at odds with findings from other research. For example, as we mentioned earlier, Bol and colleagues, in their study of a grant application call in the Netherlands, found that women received lower review scores than men but had the same probability of success [4]. In contrast, our findings reveal the opposite: there were no significant differences in review scores, but the success rate for men was higher than for women. When we disaggregate the data by domain, this gender gap appears to originate primarily from a specific field—natural resource management, which is dominated by projects related to oil and other extractive industries, arguably associated with masculinity. An implication of this finding is that scientific fields may feature nuanced gender expectations, with the perceived masculinity of a field potentially influencing participants' chances depending on their gender.

Our study suggests several avenues for future research. First, our analysis relied on regression techniques using observational data without random assignment, which introduces certain limitations. Since the composition of the National Scientific Council changes periodically (with 90% of its members replaced every three years), alternative identification strategies, such as difference-in-differences, could be employed if time-based data were available. Such approaches could address many of the challenges associated with standard regression methods, which often fail to account for the possibility that more frequent awards to members of the National Scientific Council may reflect, for instance, their superior talent or academic excellence. Employing experimental techniques or their replication with observational data would improve the internal validity of causal inferences.

Finally, future research could expand the range of independent variables, including a deeper exploration of the concept of closeness to the decision-making process. In this study, we used organizational proximity—specifically, principal investigators (PIs) sharing affiliations with council members—as a proxy for closeness to decision-making. This served as a basic indicator of potential personal connections to council members. Moving forward, examining co-authorship and citation networks could provide valuable new insights.

## Supporting information

**S1 Table. Summary statistics of the main explanatory variables.**
(DOCX)

**S2 Table. Additional OLS regressions for score variable.** This table compares an ordinary OLS regression for Score variable (all predictors included) with a model with standard errors clustered by domains (column two), a model with random effects for PI, region, and domain, and an OLS regression with truncated residuals (tobit).
(DOCX)

**S3 Table. Additional ordinal regressions for score variable.** This table presents two ordinal regression for Score variable (all predictors included). The first model takes all the possible values as ordered levels. The second model breaks original scores into 5 point intervals.
(DOCX)

**S4 Table. Additional logistic regressions for win variable.** This table presents four logistic regressions for Win. The first model is a logistic regression with robust standard errors. The second and third modes have clustered standard errors. The last model is a random effect (intercepts) model at the level of domains and regions. A model with a random intercept at the level of PI did not converge, nor did models with random slopes (at any level of grouping).
(DOCX)

**S5 Table. Lasso regressions.** This table compares the full logistic model with two lasso regressions (at different levels of lambda). The levels of lambda are chosen based on glmnet internal criteria. Made in glmnet R package [49,52].
(DOCX)

**S1 Fig. Model diagnostics for OLS regression.** The dependent variable is Score, all the available explanatory variables are used.
(PDF)

**S2 Fig. Lasso regressions.** GLMNET's selected lambda values. Made in glmnet R package [49,52].
(PDF)

**S3 Fig. ROC curves for different models.** The dependent variable is winning a grant. The diagram shows overall accuracy of the both parametric and non-parametric models.
(PDF)

**S4 Fig. Predictive performance of different models.** Logistic regressions tend to detect winning projects more often and are more sensitive than non-parametric models (fewer type II errors). The random forest predicts negative cases better because it is more conservative. For example, it predicts winning a grant only for 11% of applications. However, it does so, its more precise than other models. Logit regression and XGBoost tend to be more optimistic (they predict success for 14% of applications). All of these predicted success rates are still below the actual one (25%).
(PDF)

## Acknowledgments

We thank Alexey Knorre, Sylvan Perlmutter, and Dmitriy Serebrennikov for valuable feedback on earlier drafts of the manuscript. All mistakes are our own.

## Author contributions

**Conceptualization:** Darkhan Medeuov, Adil Rodionov.

**Data curation:** Kamilya Rodionova, Zhaxylyk Sabitov, Adil Rodionov.

**Formal analysis:** Darkhan Medeuov, Adil Rodionov.

**Visualization:** Darkhan Medeuov.

**Writing – original draft:** Darkhan Medeuov, Kamilya Rodionova, Zhaxylyk Sabitov, Adil Rodionov.

**Writing – review & editing:** Darkhan Medeuov, Adil Rodionov.

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
