## [Decision Letter · Decision Letter 0]

PONE-D-24-32179Negotiating science funding: The interplay of merit, bias, and administrative discretion in grant allocation in KazakhstanPLOS ONE

Dear Dr. Medeuov,

Thank you for submitting your manuscript to PLOS ONE. After careful consideration, we feel that it has merit but does not fully meet PLOS ONE’s publication criteria as it currently stands. Therefore, we invite you to submit a revised version of the manuscript that addresses the points raised during the review process.

**Please consider carefully all suggestions of the two reviewers. Both are positive and constructive about your work. From my part, I think that if the data of your paper cannot be made available properly anonymized in a public repository, the paper cannot be accepted for publication in PLOSONE. **

We look forward to receiving your revised manuscript.

Kind regards,

Alberto Baccini, Ph.D.

Academic Editor

PLOS ONE

**Journal Requirements:**

This research was funded by the Science Committee of the Ministry of Education and Science of the Republic of Kazakhstan [grant number AP13068350].

3. In the online submission form, you indicated that Data cannot be shared publicly because it contains PI's full names. The data underlying the results presented in the study are available from darkhan.medeuov.personal@gmail.com upon request.

4. We note you have included a table to which you do not refer in the text of your manuscript. Please ensure that you refer to S1 Table in your text; if accepted, production will need this reference to link the reader to the Table.

Reviewers' comments:

Reviewer's Responses to Questions

**Comments to the Author**

1. Is the manuscript technically sound, and do the data support the conclusions?

Reviewer #1: Partly

Reviewer #2: Yes

2. Has the statistical analysis been performed appropriately and rigorously? 

Reviewer #1: Yes

Reviewer #2: Yes

3. Have the authors made all data underlying the findings in their manuscript fully available?

Reviewer #1: Yes

Reviewer #2: No

4. Is the manuscript presented in an intelligible fashion and written in standard English?

Reviewer #1: Yes

Reviewer #2: Yes

5. Review Comments to the Author

**Reviewer #1:** The paper analyses the mechanisms of selection of public funding for research projects in different fields of analysis in Kazakhstan. the study contributes to the recent research evaluation literature by proposing a different perspective from most published analyses that focus mainly on Anglo-Saxon and/or European countries. The empirical analysis of the grant funding competition held in 2017 in Kazakhstan highlights the subjective nature of the selected research projects' quality concept that unmasks selection mechanisms based on homophily and gender discrimination.

The econometric analysis is rich and interesting; however, the article needs more effort in outlining conclusions that can provide policy recommendations from the results obtained so that an improvement in public research funding selection processes can be proposed. Which selection mechanisms could correct the anomalies found? For instance, in order to correct possible gender biases in the evaluation, some institutions have proposed a quota system with an ex-ante fixed percentage of funded projects for IPs belonging to the minority sex. Could this be a feasible way forward for the analysed context? To make the paper suitable for publication I would recommend expanding and supplementing the conclusions with policy recommendations.

Furthermore, concerning the literature review, I recommend supplementing the background section with literature about homophily in research, such as:

Santos, J.M., Horta, H. & Feng, S. Homophily and its effects on collaborations and repeated collaborations: a study across scientific fields. Scientometrics 129, 1801-1823 (2024). https://doi.org/10.1007/s11192-024-04950-3

Zhou, S., Chai, S. Richard & Freeman, B. Gender homophily: In-group citation preferences and the gender disadvantage, Research Policy, 53 (1): 104895 (2024), ISSN 0048-7333,

Regarding the econometric analysis:

- The variable PI's sex is described as ‘male is the reference category’, however in the tables, the variable is defined as ‘female’ implying that instead, female is the reference category

- To complete the gender analysis, I would propose to include interaction terms in the logit model, for example, female#score and female#Hindex

- In the current version, the cross-validation section does not add any insight into the analysed phenomenon. I would suggest moving the section to the appendix or otherwise better explain how the results of the random forest and XGBoost models can complement the results obtained with the parametric methods in the previous sections.

I hope that these comments will be useful for authors.

**Reviewer #2:** This was an interesting article to read and I have an overall positive opinion of it. I particularly liked the specificity to the institutional context of Kazakhstan: as is pointed out by the introduction, studying contexts beyond the U.S. and Western Europe has special value from a comparative perspective and has special value for understudies geographies.

I have three main comments and then a list of smaller remarks. My three main comments are (1) a reflection on the role of reviewer scores; (2) a remark about the data sharing statement; (3) a complaint about the use of judgmental language in the article.

Beginning with the first point on reviewer scores. As is pointed out in the front-end of the paper, oftentimes a single negative review is all it takes for a grant to be declined. In general, any reviewer disagreement may result in rejection: seeing discordant opinions, whomever takes the final funding decisions (e.g. the panel) may simply decide that the project is too risky or controversial, and reject it (see e.g. Lane et al. 2022, https://doi.org/10.1287/mnsc.2021.4107 ). I think this might be of consequence for how review scores are operationalized in the analyses. Let me make an example to illustrate my point: two proposals are competing for the same grant and have been reviewed by the same set of reviewers. Proposal A received the scores {7, 7, 7, 7}. Proposal B has received the scores {8, 8, 8, 4}. Both proposals have the same average score of 7: this means that, if we use the average score as predictor, we would expect A and B to have the same chances of being funded. However, reviewers are in consensus for proposal A, and in (strong) disagreement for proposal B. What happened is that one of the reviewers of B may have identified problems that the other reviewers have not noticed. The panel who is tasked with the final decision might think that proposal B is therefore riskier than proposal A, and ultimately choose B over A. This goes to show that reviewer agreement, according to theory, may be as good a predictor of funding success as the average score. Hence my question: have you considered estimating the models using some measure of disagreement (e.g. the standard deviation of scores) as fixed effect in place of or in addition to the average score? Do the results say anything new?

My second main comment is about the data stamen that “Data cannot be shared publicly because it contains PI's full names”. And then “The data

underlying the results presented in the study are available from

[email] upon request”. Then is the data available or not available for secondary scrutiny? If it is available, it should be published to a persistent repository and made Findable, Accessible, Interoperable and Reusable (FAIR). If this is not possible, then how could it be made available “upon request”? Note also that, if the identifiability of PIs is the only obstacle to publishing the data, perhaps PI names can be replaced with anonymized IDs.

My third main point is about problematic language use throughout the article. For instance, I strongly recommend toning down the statement on line 87: “disciplines like sociology, where some even describe the field as a bunch of feuding scientific gangs”. The issue here is the unusually strong negative wording to single out and characterize as problematic a very large – and, yes, diverse – research field and research community. For full disclosure, I am a sociologist myself, and I recognize this might make me biased. But I would find this wording strangely off even if it was directed at other fields that are even more divided into competing schools of thought. I understand that this strong and stigmatizing wording is attributed to Scheff (1995). However, quoting it instead of more neutrally conveying the same message amounts to an endorsement of Scheff’s words from 29 years ago. Note also that PlosOne is an inherently interdisciplinary journal, and I would think and hope that remarks such as this would not land well here.

The same applies to the text at the end of page 4 ,inclusive of footnote 1: “Physics, for example, often serves as a model case of a stable-core discipline, while Sociology is the prime example of the opposite [1] Mostly because of sociologists’ fondness of self-deprecation”. I am not sure where these statements come from. They do not seem grounded in facts and frankly I find them quite offensive. Because I belong in the offended community here, I will leave the decision on their appropriateness to the editor. As for how to constructively make your point without dismissively trashing an entire discipline and alienating your readership, perhaps you can shape your argument along the following lines. Established norms on what endeavors, theories, methods or tools are more valuable than others [what you call ‘pecking order’ or hierarchy] vary across scientific fields and subfields. And some fields and subfields may have more shared and established norms than others [what you call ‘cohesion’]. Note that the angle I’m proposing serves two purposes: (1) it tries to deliver your argument without making disputable value judgements on specific fields of science. Second, (2) it avoids relying on the concepts of cohesion and hierarchy which are multifaceted and hard to define, replacing them with a more ‘operative’ definition (hierarchy = shared understanding that some theories or methods are better than others; cohesion = the scholarly community agrees on the hierarchy). The following paragraph (lines 179-185) needs to be similarly amended.

Another example of problematic wording is the naming of the variable “fake”, the binary variable that captures whether a PI had published in Scopus-(de)listed journals. The word “fake” has, again, value judgment that may be contested. A PI might have published an excellent article in a predatory journal, or in a journal that has adopted predatory practices at a later date. This per se does not automatically make the PI’s paper in that journal “fake” nor “bad” in any way. Consider renaming this variable into something more neutral / judgement-free, e.g. “Scopus-delisted publication” or similar.

The next batch of comments is about the method and specifically the operationalization of different variables.

The description of the evaluation scales on page 7 explains that the criterial scores are graded on a scale from 0 to 9. Since the direction of this scale is directly relevant for interpreting the results, the text should also explicitly say whether higher values represent better score (thus, 9 is the best grade) or the other way round (0 as the best grade).

About the variable “PI’s formal degree”. Based on your description, It seems to me that this variable captures a cohort effect rather than simply the amount of accumulated “social capital”, as you put it. Younger academics (i.e. those with a PhD), I would expect, are at a disadvantage because of a variety of effects – they have had fewer chances to accumulate previous funding; they have had less time to accumulate good and well-cited publications; and sure, they would have had less time to form those social ties that may favor them in a national competition (social capital). Separating younger and older academics there are also institutional changes that only the older generation has lived through: this might have had an impact on how they work and what they decide to work on. These and other effects can all be lumped together under the concept of “cohort effect”. If you agree, please revise the name and description of this variable.

About the variable “previous success”: this encodes whether the PI has won a call in the previous call in 2014. Note that PI might have acquired funding from other calls or other funders as well. If you agree, my advice is to make it explicit that this variable will have some (or even many?) false negatives, meaning that some/many PIs will be coded as “previous success = FALSE” even though they had previous success in calls other than the one in 2014. You may try to claim that this is a conservative estimate of previous success that seeks to capture exactly what you are interested in – previous success but also previous familiarity with what you call the “rules of the game” in science funding in Kazakhstan.

The description of the variable “PI’s sex” uses interchangeably gender (man, woman) and sex (male, female). My general advice is to choose one of the two consistently and to name this variable accordingly. The standard, to my understanding, is to use gender: hence my recommendation to choose gender if applicable. Please check if PlosOne has specific author guidelines for talking about sex, gender, and inclusive language. On the other hand, I can imagine good reasons for using sex (male/female) instead. For example, applicant forms might have explicitly asked for the biological sex of applicants rather than their gender: in this case it would make sense to use the same categories as they appear on the form, and to add a footnote or a comment to this effect. I am not familiar with the funding landscape of Kazakhstan and I cannot speak Kazakh or Russian, so I will leave it for you to determine whether this is the case.

Table 1 provides descriptive statistics for some of the variables. I would be advisable to show the distribution of the other variables as well, possibly in an appendix.

I close with more minor comments:

What I miss form the conclusion/discussion is a comment on the potential (?) generalizability of these results to other contexts. For example, knowing little about the subject I would naively think that other post-Soviet institutions have undergone similar policy changes and social processes. Do we know of – or should we expect – similar dynamics at play in other post-soviet states?

Gender bias in science funding is quite controversial. Several studies report gender differences in funding success and grant sizes (Bornmann, Mutz, and Daniel 2007 https://doi.org/10.1016/j.joi.2007.03.001 ; McAllister, Juillerat and Hunter 2016 https://doi.org/10.1038/529466d ; Burns et al. 2019 https://doi.org/10.1371/journal.pmed.1002935 ; Witteman et al. 2019 https://doi.org/10.1016/S0140-6736(18)32611-4 ). However, a couple of other studies find the opposite (Mutz, Bornmann and Daniel 2012 https://doi.org/10.1027/2151-2604/a000103 ; Yip et al. 2020 https://doi.org/10.1057/s41599-020-00656-y ). Perhaps you could explicitly link your results to this stream of literature by showing yet another domain where there is evidence for a gender effect.

The first paragraph of the introduction can be sharpened in a couple of places where the current phrasing is not quite precise. One issue is with the sentence “While merit is often cited as the ideal criterion […]”: cited by whom? Also note that there other criteria that some funders and scholars may deem to be even more important than merit, such as scientific relevance or the potential for impact. An easy workaround here could be to either provide some references or to tone down this statement by rewriting it along these lines: “Merit is certainly an important criterion; however, […]”.

The other sentence in the introduction that can be sharpened is the one that reads “since science depends on public money, […]”. The issue is that not all science depends (exclusively or at all) on public money. Here, too, the fix could be quite straightforward: “since science often relies on public funds, […]”.

Information on the two stages of review presented across the second and third paragraph comes across as somewhat redundant. Please consider merging the two explanations.

Top of the background and theory section: “scientific peers are the most suitable juries because their internal rankings predict future scientific output the best”. This is presented as a fact – if it is, it begs for supporting references. If it is an opinion, as I think it is, it should be phrased more carefully.

The bottom of the “Context” section describes the fallout and public scandal that followed the publication of winners’ list the 2017 call. Some of it – the description of an open letter by scientists, the involvement of a former President and their ministry – is not supported by sources as I would expect (e.g. references to the open letter and newspaper articles outlining what is described).

Line 417: “a PI could lead at least two projects”. I presume this is an error, and the text should actually read “a PI could lead *at most* two projects”.

Regression tables or their caption should remind the reader what the dependent variable is.

6. PLOS authors have the option to publish the peer review history of their article (what does this mean?). If published, this will include your full peer review and any attached files.

Reviewer #1: No

Reviewer #2: No

---

## [Author Response · Author response to Decision Letter 1]

13 Dec 2024

Thank you very much for the thorough and valuable feedback, as well as for the opportunity to revise and resubmit our manuscript to PLoS ONE. We have divided our response into three parts: (1) comments from the Editors (2) comments from Reviewer 1, and (3) comments from Reviewer 2.

Response to the Editor

Dear Prof. Baccini,

Thank you very much for your valuable feedback.

We made every effort to carefully respond the reviewers' comments and address all their questions thoroughly. The argument has been streamlined, and the article's contributions have been strengthened. The conclusion has been thoroughly rewritten and expanded to include detailed recommendations.

Additionally, we closely adhered to the journal's requirements. We included the statement, "The funders had no role in study design, data collection and analysis, the decision to publish, or the preparation of the manuscript," in the financial disclosure section. The data have been anonymized and uploaded to a public repository in compliance with the journal's guidelines. The manuscript has been reviewed for style according to PLOS ONE's requirements, and we have made every effort to correct grammatical mistakes and inconsistencies.

Response to Reviewer 1

We thank Reviewer 1 (R1) for their extremely helpful comments and suggestions. The feedback helped us to strengthened and clarify the paper's arguments.

Let us move on to specific comments.

R1 suggested enhancing the conclusion of our article by incorporating policy recommendations based on the received results. These recommendations could prove valuable in improving the selection processes. Specifically, R1 wrote:

however, the article needs more effort in outlining conclusions that can provide policy recommendations from the results obtained so that an improvement in public research funding selection processes can be proposed. Which selection mechanisms could correct the anomalies found? For instance, in order to correct possible gender biases in the evaluation, some institutions have proposed a quota system with an ex-ante fixed percentage of funded projects for IPs belonging to the minority sex. Could this be a feasible way forward for the analysed context? To make the paper suitable for publication I would recommend expanding and supplementing the conclusions with policy recommendations.

We have added the following passage to the conclusion section (lines 641-670),

Based on the findings, several policy recommendations can be made to improve the efficiency of grant allocation. The first recommendation addresses gender inequality. It is crucial to determine whether this inequality is a recurring issue in grant competitions or specific to certain calls or domains. As our data suggest gender bias in the 2017 call was more pronounced in the domain of Natural Resource Management, mostly composed of projects related to oil industry.

If gender inequality proves to be a consistent problem across domains, implementing temporary quotas could help make up for biases. Previous research has shown that, while Kazakhstani academia has achieved quantitative gender parity — evidenced by the growing number of women obtaining scientific degrees — significant gender inequality persists [49]. In this context, such measures could be highly effective.

The second key policy recommendation concerns the members of the National Scientific Council (NSC). As decision-makers, they were also involved in the competition, which created a clear conflict of interest. The Ministry of Science and Higher Education has already taken steps to address this issue. In subsequent competitions, NSC members no longer have the final say in determining which projects receive funding; instead, they score projects alongside external reviewers. However, further in-depth research is needed to clarify the councils role, evaluate their contributions to the final decisions, and open the "black box" of communication between council members during the negotiations. For example, one mechanism that may contribute to gender bias is gender homophily coupled with disproportionate number of men in research councils. In other words, possible interventions may focus not on the outcome of the competition ( for example, gender quotas for winning projects), but on gender composition of the councils (gender quotas for council members).

It would be misguided to assume that the issue with research councils is unique to Kazakhstan. Similar cases have been observed in European contexts, where selection committees sometimes disregard review scores [6] or exhibit self-serving bias [41]. Therefore, it is important not to exoticize the Kazakhstani case but to develop a unifying framework for addressing issues of gender inequality and the balance between external and internal expertise.

R1 recommended supplementing the background section with literature about homophily in research. In particular, he/she suggested the following sources of literature:

Santos, J.M., Horta, H. & Feng, S. Homophily and its effects on collaborations and repeated collaborations: a study across scientific fields. Scientometrics 129, 1801-1823 (2024). https://doi.org/10.1007/s11192-024-04950-3

Zhou, S., Chai, S. Richard & Freeman, B. Gender homophily: In-group citation preferences and the gender disadvantage, Research Policy, 53 (1): 104895 (2024), ISSN 0048-7333

Thank you very much for these useful sources of literature. We have updated the manuscript by quoting these sources in the theory section.

We have augmented Background and Theory section with the following text (p. 4, lines 122-134):

Finally, it is crucial to recognize that non-meritocratic factors, such as an applicant's gender, ethnicity, or race, can significantly impact the decision-making process. For example, gender often serves as a primary characteristic for homophily in scientific collaborations, meaning that individuals tend to form partnerships and networks with those of the same gender. As a consequence, not only collaboration networks but also citation networks show systematic gender homophily [33; 34]. Research also shows systematic penalties for women in both hiring and grant funding [4; 8].

We also cited these research when describing our independent variable - PI’s gender.

The variable PI's sex is described as ‘male is the reference category’, however in the tables, the variable is defined as ‘female’ implying that instead, female is the reference category

We apologize for the confusion. In the regression tables, we write “Female PI” to indicate that it is a coefficient reflecting female PI’s chances relative to the reference category - “male PI”. To avoid confusion, we have added clarifications in the captions.

To complete the gender analysis, I would propose to include interaction terms in the logit model, for example, female#score and female#Hindex

We have incorporated interaction terms in the logit models in the table 4 (page 16)

In the current version, the cross-validation section does not add any insight into the analysed phenomenon. I would suggest moving the section to the appendix or otherwise better explain how the results of the random forest and XGBoost models can complement the results obtained with the parametric methods in the previous sections.

We agree that in its current form cross-validation section does not add much insight. Our aim, however, was to illustrate underperformance of more flexible non-parametric models. Yet again, we agree that cross-validation has minimal impact on the main findings; therefore, we have relocated this analysis to the supplementary materials.

We appreciate R1’s thoughtful remarks on our manuscript and helpful suggestions.

Response to Reviewer 2

We are grateful to Reviewer 2 (R2) for immensely useful comments and feedback on our paper.

R2 provided three main comments, which we will address first. The first comment pertains to a reflection on the role of reviewer scores. R2 writes,

“Beginning with the first point on reviewer scores. As is pointed out in the front-end of the paper, oftentimes a single negative review is all it takes for a grant to be declined. In general, any reviewer disagreement may result in rejection: seeing discordant opinions, whomever takes the final funding decisions (e.g. the panel) may simply decide that the project is too risky or controversial, and reject it (see e.g. Lane et al. 2022, https://doi.org/10.1287/mnsc.2021.4107 ). I think this might be of consequence for how review scores are operationalized in the analyses. Let me make an example to illustrate my point: two proposals are competing for the same grant and have been reviewed by the same set of reviewers. Proposal A received the scores {7, 7, 7, 7}. Proposal B has received the scores {8, 8, 8, 4}. Both proposals have the same average score of 7: this means that, if we use the average score as predictor, we would expect A and B to have the same chances of being funded. However, reviewers are in consensus for proposal A, and in (strong) disagreement for proposal B. What happened is that one of the reviewers of B may have identified problems that the other reviewers have not noticed. The panel who is tasked with the final decision might think that proposal B is therefore riskier than proposal A, and ultimately choose B over A. This goes to show that reviewer agreement, according to theory, may be as good a predictor of funding success as the average score. Hence my question: have you considered estimating the models using some measure of disagreement (e.g. the standard deviation of scores) as fixed effect in place of or in addition to the average score? Do the results say anything new?”

This is certainly a very important commentary that highlights the issue of reviewer agreement. We would be glad to test this idea; however, the data are structured in a way that makes it impossible. Specifically, we only have access to the average rating from three reviewers, not the individual ratings. During the data collection phase, we attempted to request data on each individual reviewer’s scores, but our request was denied. For reasons that remain unclear, these scores are considered classified information. Consequently, only the average review score could be used in this study.

R2's second main comment was about data sharing statement, they wrote

“My second main comment is about the data stamen that “Data cannot be shared publicly because it contains PI's full names”. And then “The data

underlying the results presented in the study are available from

[email] upon request”. Then is the data available or not available for secondary scrutiny? If it is available, it should be published to a persistent repository and made Findable, Accessible, Interoperable and Reusable (FAIR). If this is not possible, then how could it be made available “upon request”? Note also that, if the identifiability of PIs is the only obstacle to publishing the data, perhaps PI names can be replaced with anonymized IDs.”

We have anonymized our dataset and uploaded it to a public repository.

R2's third main comment concerns a complaint about the use of judgmental language.

First, we apologize for the careless choice of words. Most of the authors are sociologists themselves, and we may have become overly critical in our self-reflection. We did not mean to devalue the work of fellow sociologists. We have rewritten the related paragraphs in a more professional and neutral tone. And we appreciate the restatement suggested by the reviewer.

In particular, we have either completely deleted or rewritten paragraphs in lines 78-87 which reads now,

Partly, peer reviews can appear inconsistent due to factors such as reviewer fatigue, vague evaluation criteria (e.g., significance or novelty), or a lack of reviewing experience [14, 24]. However, disagreements between reviewers also stem from the inherent ambiguity of merit. Scientific communities may agree on what constitutes quality science and who among them comes closest to doing it, yet the literature indicates that consensus is not always reached. Many scientific fields have controversial topics that split communities into what are often called schools of thought [12, 25]. While the degree of (perceived) fragmentation vary across disciplines [25–27], sociological studies of scientific knowledge show that the nature of scientific fact can be, and often is, contested even in the most stable of hard sciences like physics or chemistry [28–31].

We have also deleted a footnote on the page 4;

and we have rewritten the paragraph in the lines 179-185 (now it is lines 172-183), which now reads,

Scientific disciplines vary in the extent to which they agree on preferred endeavors, theories, methods, or tools. Some disciplines feature institutional structures that impose a higher degree of homogeneity on the education and training required for students to begin their scientific careers [37, 38]. Others disciplines exhibit greater diversity in their epistemic styles [2]. We argue that the level of epistemic diversity may correlate with the stability of evaluation criteria. Specifically, disciplines with more homogenous epistemic styles may tend to have more fixed sets of criteria for distributing resources (e.g., grants, academic positions). This suggests that in fields with low epistemic diversity, science managers may be more inclined to rely on external experts, whose authority helps reduce uncertainty (and potentially shift responsibility). Conversely, disciplines characterized by more diverse epistemic cultures often, by having more flexible evaluation criteria, allow administrators to exercise greater discretion in their decisions.

Reviewer 2 also writes,

The description of the evaluation scales on page 7 explains that the criterial scores are graded on a scale from 0 to 9. Since the direction of this scale is directly relevant for interpreting the results, the text should also explicitly say whether higher values represent better score (thus, 9 is the best grade) or the other way round (0 as the best grade).

We agree and we have explicated the direction of the scale in the text, now it reads (lines 308-310),

each application had been graded by three independent anonymous experts based on four criteria. Each criteria ran on 0 to 9 scale, where 9 is the best grade, and consisted of 2-3 categories with a set of cue questions.

Reviewer 2 writes,

about the variable “PI’s formal degree”. Based on your description, It seems to me that this variable captures a cohort effect rather than simply the amount of accumulated “social capital”, as you put it. Younger academics (i.e. those with a PhD), I would expect, are at a disadvantage because of a variety of effects – they have had fewer chances to accumulate previous funding; they have had less time to accumulate good and well-cited publications; and sure, they would have had less time to form those social ties that may favor them in a national competition (social capital). Separating younger and older academics there are also institutional changes that only the older generation has lived through: this might have had an impact on how they work and what they decide to work on. These and other effects can all be lumped together under the concept of “cohort effect”. If you agree, please revise the name and description of this variable.

We agree that a cohort effect would be a better description; we have revised the the corresponding paragraph which now reads (lines 374-382),

PI’s formal degree (categorical, 3 levels). This variable has three categories: Candidate of Science (reference level), Doctor of Science, and PhD. Before joining the Bologna Process in 2011, Kazakhstan followed the Soviet two-tier system of scientific degrees: Candidate of Science (first tier) and Doctor of Science (second tier). Since the 2000s, this system has been replaced by the single PhD degree. Because we do not have data on the year of degree award, this variable may capture cohort effects across different generations of academics. It likely reflects variation in the accumulation of social capital throughout participants’ careers, as well as differences in institutional experience, both of which ma

---

## [Decision Letter · Decision Letter 1]

PONE-D-24-32179R1Negotiating science funding: The interplay of merit, bias, and administrative discretion in grant allocation in KazakhstanPLOS ONE

Dear Dr. Medeuov,

Thank you for submitting your manuscript to PLOS ONE. After careful consideration, we feel that it has merit but does not fully meet PLOS ONE’s publication criteria as it currently stands. Therefore, we invite you to submit a revised version of the manuscript that addresses the points raised during the review process. A statistical reviewer highlighted major issues in your paper. Please consider carefully all suggestions.

 Please submit your revised manuscript by Mar 23 2025 11:59PM. If you will need more time than this to complete your revisions, please reply to this message or contact the journal office at plosone@plos.org. Please include the following items when submitting your revised manuscript:

We look forward to receiving your revised manuscript.

Kind regards,

Alberto Baccini, Ph.D.

Academic Editor

PLOS ONE

Reviewers' comments:

Reviewer's Responses to Questions

**Comments to the Author**

1. If the authors have adequately addressed your comments raised in a previous round of review and you feel that this manuscript is now acceptable for publication, you may indicate that here to bypass the “Comments to the Author” section, enter your conflict of interest statement in the “Confidential to Editor” section, and submit your "Accept" recommendation.

Reviewer #2: All comments have been addressed

Reviewer #3: (No Response)

2. Is the manuscript technically sound, and do the data support the conclusions?

Reviewer #2: Yes

Reviewer #3: Partly

3. Has the statistical analysis been performed appropriately and rigorously? 

Reviewer #2: Yes

Reviewer #3: No

4. Have the authors made all data underlying the findings in their manuscript fully available?

Reviewer #2: Yes

Reviewer #3: No

5. Is the manuscript presented in an intelligible fashion and written in standard English?

Reviewer #2: Yes

Reviewer #3: Yes

6. Review Comments to the Author

Reviewer #2: This was a strong paper to begin with, and now the authors have thoroughly and thoughtfully addressed all the comments I raised. I am happy to recommend acceptance and wish the authors good luck.

Reviewer #3: The manuscript addresses an interesting topic. I really enjoyed reading the paper and the empirical idea could be applied to different countries, as the research question developed here is common worldwide. Here I focus on the employed statistical methods and comment on these mainly. I feel that the empirical discussion is rather well-done, but there is room for improvements on the methods. My comments follow.

1. I warmly thank the author for sharing the data. It would be nice to have the code used to obtain the results as well. This would ensure the replication of the results.

2. Up to what I understand, the scores are discrete and as such should be treated. Here instead the outcome is considered continuous and linear regression is applied. I cast some doubts on the appropriateness of the methods. Please, provide evidence that the Gauss-Markov assumptions are met. This is crucial and mandatory to ensure the reliability of the inferential results. Indeed, I suggest to go for a generalized linear model instead, properly accounting for the discrete nature of the outcome variable.

3. The data have a clear hierarchical structure: there are repeated measurements for some PIs, there might be a judge-effect, and also the field may have an impact. Overall, what I am saying is that there might be unobserved heterogeneity at different levels of the hierarchy which may affect the inferential results. I strongly suggest to include random effects at the different levels of the hierarchy to account for such heterogeneities. Furthermore, relying on Aitkin, M. (1999). A general maximum likelihood analysis of variance components in generalized linear models. Biometrics, 55(1), 117-128 it would be interesting to consider discrete random effects leading to a finite mixture model, i.e. they allow for clustering PIs, etc. Functions are available in R (npmlreg package), Stata (gllamm), and other software.

4. Budget constraints on the probability of winning may also be an issue. I have to admit that I do not the specific context, but in my country the budget is allocated field-wise to ensure that all the fields would receive some funds. Moreover, there might be the case where two projects share the same grade but one only is financed due to lack of economic resources.

5. It would be nice to see how well the logistic model (again I suggest to move to a logistic mixed model instead) is able to correctly identify the funded projects. Specificity and sensitivity could be added to the analysis, for example.

6. As a plus, I suggest to consider variable selection techniques. There are many independent variables in the model and most of them do not have any explanatory power. Thus, procedures like lasso, ridge or elastic net may help in identifying the main determinants.

7. PLOS authors have the option to publish the peer review history of their article (what does this mean?). If published, this will include your full peer review and any attached files.

Reviewer #2: No

Reviewer #3: No

---

## [Author Response · Author response to Decision Letter 2]

10 Apr 2025

We thank PloS ONE editorial team as well as the reviewers for their valuable feedback and the opportunity to refine our manuscript. Below are our responses to the reviewers’ comments.

Response to Reviewer 2.

We cordially thank Reviewer 2 for their previous comments which really helped us to revise the manuscript. We also deeply appreciate their approval of our work.

Response to Reviewer 3.

We are very grateful to Reviwer 3 for their detailed methodological comments. We address the comments as follows,

Reviewer 3’s wrote that it would be nice if we share the code used to conduct analysis.

We totally agree and have added the replication code to the repository (at https://dataverse.harvard.edu/dataset.xhtml?persistentId=doi:10.7910/DVN/8UZ7EY).

Reviewer 3’s wrote that,

Up to what I understand, the scores are discrete and as such should be treated. Here instead the outcome is considered continuous and linear regression is applied. I cast some doubts on the appropriateness of the methods. Please, provide evidence that the Gauss-Markov assumptions are met. This is crucial and mandatory to ensure the reliability of the inferential results. Indeed, I suggest to go for a generalized linear model instead, properly accounting for the discrete nature of the outcome variable.

We thank the Reviewer for pointing this out. The scores that characterize projects are averages of three peer reviewers’ evaluations on the scale from 0 to 36. They can take fractional values (e.g. 35.67), however they are discrete in the sense that they has a finite support (their fractional part can only be .00, .33, or 0.67). In particular, score variable takes on 111 (3*37) distinct values.

The Reviewer suggested that to test Markov-Gauss assumptions. We have added model diagnostic plots to the supplement (S1_figure.pdf). The diagnostics indicate that residuals tend to align with a normal distribution up until upper quantiles where empirical quantiles tend to be closer to the mean that the theoretical expectation. This basically reflects the natural upper boundary for the score (36 points) and that the predicted values tend to underestimate real ones as they approach the upper boundary. To account for these issues, we, run additional regression models: with clustered standard errors and a tobit-regression (with residuals having truncated normal distribution). The results are included in the supplement materials (S2_table.docx).

The Reviewer also suggested using generalized linear models. We agree and we have modelled score as an ordered categorical variable. We have added the results to the supplement (S3_table.docx).

Finally, we would like to add that we model score variable not to explain its variation but rather to provide a thick description, to illustrate how its conditional mean is associated with various characteristics of the projects. We realise that in its current edition, the manuscript does not communicate it clearly, so we have added an explicit statement on the pages.

So, to sum up how did we address the reviewers comment.

In lines 494-504, we added the following paragraph

We treat the variable review score here as a continuous one, while in fact it is effectively discrete as it can take on only a finite number of values. More over, conceptually it is more sound to treat score as an ordinal variable, as its values reflect levels of academic fitness assessed by the reviewers (akin to IELTS or TOEFL scores). A more principled approach would be to model reivew score with an ordinal regression treating its values as ordered levels.

This paper, however, aims to illustrate the effect of scores on the probability of getting a grant. In other words, review score is an explanatory variable, and our OLS regressions do not aim to explain it, but rather to describe it in relation to other variables. So as not to sidetrack too much, we organize the analysis as follows. Below, in Table 2, we present the results of OLS regressions that treat review score as a continuous variable. In the supplementary materials, we run model diagnostics and alternative models that treat score as a discrete and ordinal variable.

In supporting information, we have added models diagnostics (S1_figure.pdf), and additional regression tables (S2_table.docx, S3_table.docx).

Reviewer 3 wrote that,

The data have a clear hierarchical structure: there are repeated measurements for some PIs, there might be a judge-effect, and also the field may have an impact. Overall, what I am saying is that there might be unobserved heterogeneity at different levels of the hierarchy which may affect the inferential results. I strongly suggest to include random effects at the different levels of the hierarchy to account for such heterogeneities. Furthermore, relying on Aitkin, M. (1999). A general maximum likelihood analysis of variance components in generalized linear models. Biometrics, 55(1), 117-128 it would be interesting to consider discrete random effects leading to a finite mixture model, i.e. they allow for clustering PIs, etc. Functions are available in R (npmlreg package), Stata (gllamm), and other software.

We agree. We have run models with random effects at the level of PI, region, and domain. We have also augmented our previous models with robust standard errors clustered at the domain level. In the tables 4 and 5 of the manuscript we fit models with robust standard errors. In the supporting information, we have added a table that compare reported tables with mixed-effects logistic regressions (S4_table.docx).

Reviewer 3 wrote that,

Budget constraints on the probability of winning may also be an issue. I have to admit that I do not the specific context, but in my country the budget is allocated field-wise to ensure that all the fields would receive some funds. Moreover, there might be the case where two projects share the same grade but one only is financed due to lack of economic resources.

We absolutely agree — it is likely that budget matters, in the sense that council members may explicitly consider the amount of money requested when deciding for or against a project. However, there is no publicly available information on the funding amounts requested by projects; nor is there information on domain-specific constraints. In the data at hand, this is perhaps an unobserved source of variation. We have added a paragraph acknowledging this concern in lines (646-650).

Part of this "unexplained" success rate is likely due to unobserved project characteristics. For example, some projects might be funded simply because they were relatively inexpensive. In other words, council members might take the requested budget into account when deciding for or against a project. Unfortunately, we could not find publicly available data on project budgets.

Reviewer 3 wrote

It would be nice to see how well the logistic model (again I suggest to move to a logistic mixed model instead) is able to correctly identify the funded projects. Specificity and sensitivity could be added to the analysis, for example.

We agree and we have added model performance tests (specificity and sensitivity along a set of other metrics) to the supporting information (S3_figure.pdf, S4_figure.pdf).

We have also a short cross-validaiton section (lines 624-640) that reads,

Logistic regression is a well-established statistical method to model probability of a given event. However, in terms of predictive performance, logistic regressions do not always match with more recent non-parametric predictive models such as random forest. This predictive performance, however, comes at the cost of interpretation; non-parametric models do not have a closed-form equation connecting response and predictor variables. Nevertheless, comparing logit models with more flexible non-parametric models in terms of predictive performance helps us illustrate that in our case logistic models are on par with non-parametric ones.

We estimate the performance of the models using two 10-fold cross-validations. We choose three models from the logistic regression branch, one full model specification (with all available predictors), and two models with interaction effects (score X domain and score X sex). We also select two specifications for both random forest and XGBoost models. For those non-parametric models, we fine-tune parameters over a Latin hypercube grid to optimize the receiver operator area under the curve (roc auc) and logarithmic loss (log loss). S3 Figure and S4 Figure in the supplement materials summarize our performance tests. The tests were done in Tidymodels R package [50].

Reviewer 3 wrote,

As a plus, I suggest to consider variable selection techniques. There are many independent variables in the model and most of them do not have any explanatory power. Thus, procedures like lasso, ridge or elastic net may help in identifying the main determinants.

We agree. We have run a set of lasso regressions and compated their estimates with those of regular logistic regressions. The results are added to the supporting information (S2_figure.pdf, S5_table.pdf). Also, we added the following explanation in the manuscript (lines 610-623)

Before moving on, we would like to briefly address the issue of model selection. Our models yield many non-significant coefficients, yet we retain them. From a parsimony perspective, this is suboptimal, as it would make more sense to retain only those variables that possess greater explanatory power—and thus simplify the models.

The reason for retaining non-significant coefficients is primarily conceptual. Our explanatory variables were selected based on: (a) theoretical relevance and (b) data availability. Our main goal was to test whether these variables are associated with the probability of funding. When a variable is not (for example, the h-index), it still provides valuable information, as it arguably suggests that the review score already captures what the h-index measures.

However, to demonstrate the robustness of the coefficients, we fit several LASSO regressions [49] and compare their coefficients with those obtained in our final model (excluding interaction effects). The results are presented in the Supporting Information (S2 Figure, S5 Table).

---

## [Decision Letter · Decision Letter 2]

Negotiating science funding: The interplay of merit, bias, and administrative discretion in grant allocation in Kazakhstan

PONE-D-24-32179R2

Dear Dr. Medeuov,

We’re pleased to inform you that your manuscript has been judged scientifically suitable for publication and will be formally accepted for publication once it meets all outstanding technical requirements.

Kind regards,

Alberto Baccini, Ph.D.

Academic Editor

PLOS ONE

Additional Editor Comments (optional):

Reviewers' comments:

Reviewer's Responses to Questions

**Comments to the Author**

1. If the authors have adequately addressed your comments raised in a previous round of review and you feel that this manuscript is now acceptable for publication, you may indicate that here to bypass the “Comments to the Author” section, enter your conflict of interest statement in the “Confidential to Editor” section, and submit your "Accept" recommendation.

Reviewer #2: All comments have been addressed

Reviewer #3: All comments have been addressed

2. Is the manuscript technically sound, and do the data support the conclusions?

Reviewer #2: Yes

Reviewer #3: (No Response)

3. Has the statistical analysis been performed appropriately and rigorously? 

Reviewer #2: Yes

Reviewer #3: (No Response)

4. Have the authors made all data underlying the findings in their manuscript fully available?

Reviewer #2: Yes

Reviewer #3: (No Response)

5. Is the manuscript presented in an intelligible fashion and written in standard English?

Reviewer #2: Yes

Reviewer #3: (No Response)

6. Review Comments to the Author

Reviewer #2: (No Response)

Reviewer #3: (No Response)

7. PLOS authors have the option to publish the peer review history of their article (what does this mean?). If published, this will include your full peer review and any attached files.

Reviewer #2: No

Reviewer #3: No

---

## [Editor Report · Acceptance letter]

PONE-D-24-32179R2

PLOS ONE

Dear Dr. Medeuov,

I'm pleased to inform you that your manuscript has been deemed suitable for publication in PLOS ONE. Congratulations! Your manuscript is now being handed over to our production team.

Kind regards,

on behalf of

Prof. Alberto Baccini

Academic Editor

PLOS ONE